# Novelty Detection with Augmented Localized Conformal $p$-values

## Abstract

Novelty detection is an important area of research in both statistics and machine learning. In this paper, we focus on the conditional novelty detection problem, where novelties arise from the relationship between different variables. We first adopt the conformal inference framework and propose the Augmented Localized Conformal $p$-values (ALCP), constructed by recalibrating an augmented conditional distribution estimator. This estimator efficiently captures conditional information by incorporating both calibration and test data into its kernel estimation. We show that the resulting $p$-values are valid in finite samples and can improve detection efficiency. Based on ALCP, we then develop a novel conditional novelty detection algorithm, along with a data-driven bandwidth selection method that ensures finite-sample false discovery rate (FDR) control while enhancing detection power. Both simulated and real data experiments demonstrate the advantages of the proposed ALCP approach.

## 1 Introduction

Novelty detection plays a pivotal role in numerous scientific and industrial domains. The objective of novelty detection is to identify observations that significantly deviate from the expected pattern. In fields such as finance, healthcare, industrial monitoring and scientific discovery, novelties often correspond to fraud, disease outbreaks, fatal machine errors and identification of unexpected patterns, respectively (Li et al., 2014; Röchner and Rothlauf, 2023; Del Buono et al., 2022). The ability to accurately detect such novelties not only prevents economic and social losses but also facilitates timely decision-making.

The novelty detection problem can be formulated rigorously from a statistical perspective. In novelty detection problems, we generally have access to a reference distribution $P$ or a reference dataset $\{Z_i\}_{i=1}^n$ sampled from $P$. For a test sample $Z_{n+1} \sim Q$, the target is to determine whether $P = Q$ or not. That is, to test for the hypothesis

$$\mathbb{H}_0 : P = Q, \text{ versus } \mathbb{H}_1 : P \neq Q.$$

While this characterizes the marginal pattern, there is another line of scenarios where we are more interested in the relationship between different variables of the data. That is, we need to test for **conditional novelties** (Song et al., 2007; Hong and Hauskrecht, 2015). Let $Z := (X, W)$ where $X \in \mathbb{R}^d, W \in \mathbb{R}$, conditional novelty detection aims to detect novelties regarding the conditional distribution of $W \mid X$, which is equivalent to the following hypothesis test:

$$\mathbb{H}_0 : P_{W|X} = Q_{W|X}, \text{ versus } \mathbb{H}_1 : P_{W|X} \neq Q_{W|X}.$$

For instance, in fraud monitoring, a transaction may be statistically common in general, but highly unusual for a specific user group. In a bike-sharing dataset, one would like to identify novelties of the rental bike counts which tend to peak around 8:00 and 18:00. To determine whether an observation is a novelty, it is necessary to examine the distribution of rental bike counts conditional on different time periods during the day.

## 1.1 Related works and motivation

Classical novelty detection methods can be broadly categorized into model-based approaches and machine learning-based approaches. Model-based approaches typically rely on assumptions about the data distribution and use robust statistical estimators to identify points that significantly deviate from the estimated structure (Jobe and Pokojovy, 2015; Boukerche et al., 2020). In contrast, machine learning-based approaches treat anomaly detection as a learning problem, especially suitable for high-dimensional or large-scale settings (Liu et al., 2008; Erfani et al., 2016; Hariri et al., 2019). While those existing approaches have shown substantial utility, many of them rely on specific distributional assumptions or heuristic scoring mechanisms that may lack theoretical guarantees.

In recent years, conformal prediction (Vovk et al., 2005; Lei et al., 2018; Angelopoulos et al., 2023) has become more and more popular for its ability to provide a flexible and distribution-free framework of uncertainty quantification for black-box models. Given a reference data set, conformal prediction constructs prediction intervals for new test data points with finite-sample coverage guarantees. In addition to constructing prediction intervals, the conformal prediction framework has also been applied to the marginal novelty detection problem, where the conformal $p$-value is defined to assess how well a new observation conforms to a reference distribution. (Bates et al., 2023; Jin and Candès, 2023b; Liang et al., 2024b).

To apply conformal inference to conditional novelty detection problems, Wu et al. (2025) proposed to construct the localized conformal $p$-value (LCP) for each test sample and then to determine it as a novelty if the corresponding $p$-value is rejected. The LCP incorporates localization into $p$-value construction with kernel estimation based on calibration data, enabling it to capture information about the conditional distribution $W \mid X$. However, such localization can hamper the efficiency of LCP, since the kernel weighting technique usually reduces the effective sample size (Hall et al., 1999). The LCP may even perform worse than the marginal conformal $p$-value especially in cases with very limited calibration data. Note that a large amount of test data $\mathcal{D}_u$ contains helpful information that is ignored by the LCP when constructing the $p$-value. It is worth investigating how to integrate these test samples to improve the $p$-value's efficiency. Additionally, the performance of LCP is sensitive to the choice of bandwidth $h$ in the kernel function, which could be difficult to determine in real applications.

To address these challenges, we propose a more efficient method for constructing localized conformal $p$-values, termed Augmented Localized Conformal $p$-value (ALCP). While LCP relies solely on calibration data to estimate the conditional distribution via kernel estimation, our ALCP method introduces an augmented kernel estimator that incorporates both calibration and test data, thereby enhancing the efficiency of $p$-values. However, since the test data may include a proportion of novelties that violate exchangeability assumptions, we introduce a density-ratio estimator to account for potential distributional shifts between the calibration and test datasets. This adjustment ensures robustness in the presence of such shifts. The proposed $p$-values are seamlessly integrated with the conditional calibration framework (Fithian and Lei, 2022), resulting in a novel novelty detection algorithm. This algorithm not only guarantees a finite-sample false discovery rate (FDR) but also demonstrates superior statistical power compared to existing methods. Furthermore, we propose an adaptive bandwidth selection strategy that maximizes power in the context of novelty detection, further enhancing the method's performance.

## 1.2 Our contributions

In this work, the proposed augmented localized conformal $p$-value widens the scope of localization technique and is applicable in the case with limited calibration source. Our contribution can be summarized as follows:

- We propose the ALCP by constructing kernel conditional distribution estimators with both calibration and test data and then performing a recalibration step. The proposed $p$-value is shown to be finite-sample valid and be more efficient than LCP.

- We derive a novelty detection algorithm with the ALCP. By utilizing the conditional calibration technique, we prove that our proposed algorithm ensures finite-sample FDR control. The use of ALCP helps to identify more novelties, especially in heteroscedastic scenarios.

- We provide a reliable strategy for selecting bandwidths of the kernel estimator in a data-dependent manner. With a slight modification to the testing procedure, the optimal bandwidth can be determined by an optimization step. With this adaptive bandwidth, the proposed algorithm for novelty detection can still control the FDR in finite samples.

## 2 Augmented localized conformal $p$-value

In this section, we first provide a brief review of using conformal $p$-values for novelty detection in Section 2.1. Next, we consider conditional novelties, introduce the augmented localized conformal $p$-value and establish its finite-sample validity and asymptotic efficiency in Section 2.2.

### 2.1 Recap: novelty detection with conformal $p$-value

Consider the marginal novelty detection setting, where we observe i.i.d. data $\{Z_i\}_{i=1}^n \sim P$ and a test dataset $\mathcal{D}_u = \{Z_j\}_{j=n+1}^{n+m}$ with index set $\mathcal{U} = \{n+1, \ldots, n+m\}$. Let $\mathcal{I} \subseteq \mathcal{U}$ denote the index set of inliers and $\mathcal{O} \subseteq \mathcal{U}$ denote the index sets of novelties, respectively. Our goal is to identify as many true novelties in $\mathcal{D}_u$ as possible while controlling the FDR at some pre-specified level $\alpha > 0$:

$$\text{FDR} = \mathbb{E}\left(\frac{|\mathcal{R} \cap \mathcal{I}|}{|\mathcal{R}| \vee 1}\right) \leq \alpha$$

where $\mathcal{R}$ is the set of identified novelties. Bates et al. (2023) proposes to use conformal $p$-values for novelty detection, offering a flexible and robust inference tool without requiring explicit modeling of the data distribution. The data is split into a training set $\mathcal{D}_t = \{Z_i\}_{i \in \mathcal{T}}$ and a calibration set $\mathcal{D}_c = \{Z_i\}_{i \in \mathcal{C}}$. A score function $V$, trained on $\mathcal{D}_t$, assigns a nonconformity score to each point, capturing how anomalous an observation appears under the learned model. Typical choices for $V$ include scores from one-class classifiers or distance-based methods. For one new test point $Z_{n+1}$, its score $V_{n+1}$ is compared to the calibration scores $\{V_i\}_{i \in \mathcal{C}}$, yielding a conformal $p$-value:

$$p_{n+1} = \frac{1 + \sum_{i \in \mathcal{C}} \mathbb{I}\{V_{n+1} \leq V_i\}}{|\mathcal{C}| + 1} \tag{1}$$

This $p$-value is guaranteed to be super-uniform under the null if $\mathcal{D}_c \cup \{Z_{n+1}\}$ are exchangeable.

The conformal $p$-values $\{p_j\}_{j \in \mathcal{U}}$ computed with the same calibration set are also shown to be positive regression dependent on a subset (PRDS). This allows one to directly apply the Benjamini-Hochberg (BH) procedure on $p$-values $\{p_j\}_{j \in \mathcal{U}}$ and take the rejection set $\text{BH}_\alpha(\{p_j\}_{j \in \mathcal{U}}) := \{j \in \mathcal{U} : p_j \leq \tau_\alpha\}$ as detected novelties with

$$\tau_\alpha = \max\left\{t : \frac{mt}{\sum_{j \in \mathcal{U}} \mathbb{I}\{p_j \leq t\}} \leq \alpha\right\}$$

to ensure FDR $\leq \alpha$.

### 2.2 Augmented localized conformal $p$-values

In the conditional setting, we aim to test whether a new observation $Z_{n+1} = (X_{n+1}, W_{n+1})$ is a novelty with respect to the conditional distribution $W \mid X$. Compared to the marginal setting, the key distinction lies in constructing a score function $V(X, W)$ that captures the conformity of $W$ given $X$, such as regression residuals or class-conditional likelihoods. The

conformal $p$-value is then computed using the same rank-based formula as in Equation (1), with calibration scores constructed from the conditional setting. A direct way to build $p$-value is by reverting to the original localized prediction interval in Guan (2023):

$$p^*_{\mathrm{L},n+1} = \frac{\sum_{i\in\mathcal{C}\cup\{n+1\}} H(X_i, X_{n+1})\mathbb{I}\{V_{n+1} \leq V_i\}}{\sum_{i\in\mathcal{C}\cup\{n+1\}} H(X_i, X_{n+1})}, \tag{2}$$

where $H(\cdot,\cdot)$ is a kernel function characterizing closeness between covariate values. Note that $p^*_{\mathrm{L},n+1}$ captures the effect of localized regions around the test sample as the kernel $H(X_{n+1}, \cdot)$ takes larger value on samples closer to $X_{n+1}$. However, similar to the property of localized prediction interval in Guan (2023), $p^*_{L,n+1}$ is not valid in finite samples.

Wu et al. (2025) proposed the localized conformal $p$-value as a counterpart of localized prediction intervals in Hore and Barber (2024):

$$p_{\mathrm{L},n+1} = \frac{\sum\limits_{i\in\mathcal{C}} H(X_i, \tilde{X}_{n+1})\mathbb{I}\{V_{n+1} \leq V_i\} + \xi H(X_{n+1}, \tilde{X}_{n+1})}{\sum\limits_{i\in\mathcal{C}} H(X_i, \tilde{X}_{n+1}) + H(X_{n+1}, \tilde{X}_{n+1})},$$

where $\xi$ follows the uniform distribution $\mathrm{U}[0,1]$. The $\tilde{X}_{n+1}$ is randomly sampled from the density $H(X_{n+1}, \cdot)$. By introducing the random samples $\tilde{X}_{n+1}$, this localized conformal $p$-value ensures finite-sample validity, which mirrors the approach used in the randomly localized prediction interval proposed by Hore and Barber (2024). However, as mentioned before, such localization reduces the effective sample size and makes the resulting $p$-value less efficient, particularly when the calibration set is small.

To resolve this issue, we propose an augmented localized $p$-value to improve the efficiency. Recall the original localized $p$-value in (2). We notice that $1 - p^*_{\mathrm{L},n+1}$ is exactly the kernel estimator for the conditional cumulative distribution function (CDF) of $V \mid X$ evaluated at point $(V_{n+1}, X_{n+1})$. Although not a valid $p$-value itself, $p^*_{\mathrm{L},n+1}$ is permutation-invariant with respect to $\mathcal{D}_c$. This motivates us to consider the conditional CDF estimators as new scores and recalibrate them to construct $p$-values. To be specific, we obtain those conditional CDF estimators over $i \in \mathcal{D}_c$:

$$\widehat{F}^*_i = \frac{\sum_{k\in\mathcal{C}\cup\{n+1\}} H(X_k, X_i)\mathbb{I}\{V_i \leq V_k\}}{\sum_{k\in\mathcal{C}\cup\{n+1\}} H(X_k, X_i)},$$

and then the final recalibrated $p$-value is given by

$$p_{\mathrm{rL},n+1} = \frac{\sum_{i\in\mathcal{C}} \mathbb{I}\{\widehat{F}^*_i \leq \widehat{F}^*_{n+1}\} + 1}{|\mathcal{C}| + 1}. \tag{3}$$

By the permutation-invariant property, it is easy to prove the exchangeability of $\{\widehat{F}^*_i\}_{i=1}^{n+1}$ and so the validity of $p$-value in (3). This provides an approach to construct localized conformal $p$-value given any permutation-invariant conditional CDF estimator.

As discussed in Section 1, those conditional CDF estimators $\widehat{F}^*_i$, which rely only on calibration data, may become unreliable, resulting in inefficient $p$-values. Motivated by the intuition that a more accurate estimator of the conditional CDF should lead to improved testing performance, we propose to augment the estimator by incorporating the test data. The similar technique has also been considered by Marandon et al. (2024) to boost the power of marginal novelty detection. As we will show in this section, its application in the conditional case in non-trivial and requires more careful design of the methodology.

For each $j \in \mathcal{U}$, denote $\mathcal{D}_{c,j} = \mathcal{D}_c \cup \{Z_j\}$ and $\mathcal{D}_{u,j} = \mathcal{D}_u \backslash \{Z_j\}$ with corresponding index sets $\mathcal{C}_j = \mathcal{C} \cup \{j\}$ and $\mathcal{U}_j = \mathcal{U} \backslash \{j\}$. The augmented conditional distribution estimator at $(V_j, X_j)$ for $j \in \mathcal{U}$ can then be defined as follows:

$$\widehat{F}_j(v, x) = \frac{\sum\limits_{i\in\mathcal{C}_j} H(X_i, x)\mathbb{I}\{V_i \leq v\} + \omega \sum\limits_{k\in\mathcal{U}_j} H(X_k, x)\widehat{r}(V_k, X_k)\mathbb{I}\{V_k \leq v\}}{\sum\limits_{i\in\mathcal{C}_j} H(X_i, x) + \omega \sum\limits_{k\in\mathcal{U}_j} H(X_k, x)\widehat{r}(V_k, X_k)}. \tag{4}$$

The $\widehat{r}(v, x)$ is an estimator of the conditional density ratio $\mathrm{d}Q_{V|X}/\mathrm{d}P_{V|X}(v, x)$ between the conditional distribution $V \mid X$ of calibration data and test data (Tibshirani et al., 2019) and is introduced to cope with the distribution shift between the calibration and test data since the latter may contain samples not following the null distribution. In practice, it can be estimated by fitting binary classifiers to distinguish $\{V_i\}_{i \in \mathcal{T}}$ and $\{V_j\}_{j \in \mathcal{C} \cup \mathcal{U}}$. We also introduce a weight parameter $\omega \in [0, 1]$ to balance the contribution of the test data in the estimator. If $\omega = 0$, the estimator $\hat{F}_{n+1}(v, x)$ is identical to $1 - p^*_{L,n+1}$. Generally, we can take a larger $\omega$ when a more accurate $\widehat{r}$ can be obtained since the augmented part in the estimator could be reliable in such case.

**Remark 1.** *When $\widehat{r}$ is perfectly estimated, it could serve as an efficient score function for conditional novelty detection. However, this is extremely difficult since it is equivalent to distinguish two **conditional** distributions that are very similar. Since it leads to deteriorated performance in finite samples, we do not consider this score function.*

**Remark 2.** *The kernel function $H$ can be selected from any commonly used kernel functions in kernel density estimation. In this paper, we primarily consider Gaussian kernel $H(x, x') = h^{-(d-1)}K(\|x - x'\|_2/h)$ with standard normal density function $K(\cdot)$, which is the most common in kernel methods. This is also the choice in Guan (2023) and Hore and Barber (2024).*

Given the augmented estimator, we can define the augmented localized conformal $p$-value (ALCP) for $j \in \mathcal{U}$ by a similar recalibration step:

$$p_{\mathrm{AL},j} = \frac{\sum_{i \in \mathcal{C}} \mathbb{I}\{\widehat{F}_j(V_j, X_j) \leq \widehat{F}_j(V_i, X_i)\} + 1}{|\mathcal{C}| + 1}. \tag{5}$$

Since we do not break the permutation-invariant property when constructing the augmented estimator, the ALCP is still valid in finite samples.

**Theorem 1.** *Suppose that the density ratio estimator $\widehat{r}$ is permutation invariant with respect to $\mathcal{D}_c \cup \mathcal{D}_u$. For any $j \in \mathcal{U}$ such that $\mathcal{D}_{c,j}$ are exchangeable, the corresponding ALCP satisfies*

$$\Pr(p_{\mathrm{AL},j} \leq t) \leq t, \quad 0 \leq t \leq 1.$$

Besides the exchangeable assumption on the data, the theorem only requires a permutation invariant density ratio estimator $\hat{r}$. This can be easily achieved by fitting a binary classifier to distinguish between $\mathcal{D}_t$ and $\mathcal{D}_c \cup \mathcal{D}_u$.

We also prove the augmented estimator converges faster to the true conditional distribution by leveraging the test data for kernel estimation, which serves as circumstantial evidence of ALCP's efficiency. We need the following regularity assumptions.

**Assumption 1.** *Assume that: (1) The density ratio $r(v, x)$ and the density of $P_X$ are bounded and continuous; (2) The conditional distribution of $V = V(X, W)$ satisfies*

$$\left\| F_{V|X=x}(v) - F_{V|X=x'}(v) \right\|_{\infty} \leq L \cdot \|x - x'\|_2^{\beta}$$

*for some constant $L > 0, 0 < \beta \leq 1$.*

**Theorem 2.** *Suppose Assumption 1 holds. If we take the kernel function $H(x, x') = h^{-d}K(\frac{\|x - x'\|_2}{h})$ with Gaussian kernel $K$, the estimator $\widehat{F}_j$ evaluated at any fixed data point $z = (x, w)$ satisfies*

$$|\widehat{F}_j(v, x) - F_j(v, x)| = O_p\left( \sqrt{h^{\beta} + \frac{1}{(n + \omega m)h^d}} + \frac{\omega m}{n + \omega m}h^{-d/2}\sqrt{\mathbb{E}\left[(\hat{r} - r)^2\right]} \right).$$

From the theorem, we can see that the convergence rate for the nonparametric estimation improves with the sample size $n$ replaced by $n + \omega m$. The second term accounts for the accuracy of $\widehat{r}$ and is controlled by the parameter $\omega$. This also provides theoretical support for the recommendation on how to choose $\omega$.

## 3  Conditional novelty detection algorithm

In this section, we derive conditional novelty detection algorithms by utilizing ALCP. Section 3.1 presents a detection algorithm with finite-sample FDR control. Section 3.2 introduces our data-driven bandwidth selection method, which enhances the power of the detection procedure.

### 3.1  Novelty detection

When applying ALCP to novelty detection problems, a direct application of BH procedure is insufficient, as the PRDS property no longer holds for ALCP. Instead, we employ the conditional calibration technique (Fithian and Lei, 2022), which is commonly used in conformalized multiple testing methods. This technique allows to control the FDR even when the $p$-values do not exhibit the PRDS property.

Here, we consider the functions $V, H, \widehat{r}$ and the parameter $\omega$ as fixed. For ease of notation, we denote the set of ALCP computed with calibration data $\mathcal{D}_c$, test data $\mathcal{D}_u$ and bandwidth $h$ as $\mathcal{A}(\mathcal{D}_c, \mathcal{D}_u, h)$. For each $j \in \mathcal{U}$ and $k \in \mathcal{U}_j$, define the set of auxiliary $p$-values $p_{\mathrm{AL},k}^{(j)}$ as

$$\{p_{\mathrm{AL},k}^{(j)}\}_{k \in \mathcal{U}_j} = \mathcal{A}(\mathcal{D}_{c,j}, \mathcal{D}_{u,j}, h). \tag{6}$$

Let $\mathcal{R}_j$ denote the rejection set obtained by applying the BH procedure to auxiliary $p$-values:

$$\mathcal{R}_j = \mathrm{BH}_\alpha(\{p_{\mathrm{AL},k}^{(j)}\}_{k \in \mathcal{U}_j} \cup \{0\}). \tag{7}$$

The final rejection set $\mathcal{R}$ is then defined by letting

$$\gamma^* = \max\left\{\gamma : \sum_{j \in \mathcal{U}} \mathbb{I}\{p_{\mathrm{AL},j} \le \alpha|\mathcal{R}_j|/m, \zeta_j|\mathcal{R}_j| \le \gamma\} \ge \gamma\right\}, \ \mathcal{R} = \left\{j : p_{\mathrm{AL},j} \le \frac{\alpha|\mathcal{R}_j|}{m}, \zeta_j|\mathcal{R}_j| \le \gamma^*\right\}, \tag{8}$$

where $\zeta_j \sim \mathrm{U}[0,1]$ are independent uniform random variables. The full procedure for novelty detection is summarized in Algorithm 1.

---

**Algorithm 1** Novelty detection with ALCP

**Input:** Calibration data $\mathcal{D}_c$ and test data $\mathcal{D}_u$ with index sets $\mathcal{C}$ and $\mathcal{U}$; FDR target level $\alpha \in (0,1)$; Kernel function $H(\cdot,\cdot)$ with bandwidth $h$; Score function $V$; Density ratio estimator $\widehat{r}$; Weight parameter $\omega$

1: Compute ALCP set $\{p_{\mathrm{AL},j}\}_{j \in \mathcal{U}} = \mathcal{A}(\mathcal{D}_c, \mathcal{D}_u, h)$;
2: **for** $j \in \mathcal{U}$ **do**
3:     Compute auxiliary $p$-values $\{p_{\mathrm{AL},k}^{(j)}\}_{k \in \mathcal{U}_j} = \mathcal{A}(\mathcal{D}_{c,j}, \mathcal{D}_{u,j}, h)$;
4:     Compute $\mathcal{R}_j = \mathrm{BH}_\alpha(\{p_{\mathrm{AL},k}^{(j)}\}_{k \in \mathcal{U}_j} \cup \{0\})$;
5: **end for**
6: Compute the final rejection set $\mathcal{R}$ as in (8);

**Output:** Detected novelty set $\mathcal{R}$.

---

By introducing auxiliary $p$-values and rejection set $\mathcal{R}_j$, the conditional calibration procedure decouples the dependency between the $p$-value and the rejection number, which allows for a term-wise analysis of the FDR. The following theorem shows that the detected novelty set $\mathcal{R}$ by Algorithm 2 ensures the FDR control in finite samples, which is an instantiation of the result in Fithian and Lei (2022) with conditional sets specifically designed in our procedure.

**Theorem 3.** *Suppose $P_X = Q_X$ and the density ratio estimator $\widehat{r}$ is permutation invariant with respect to $\mathcal{D}_c \cup \mathcal{D}_u$. The final rejection set $\mathcal{R}$ output by Algorithm 1 ensures* FDR $\le \alpha$.

### 3.2  Data-driven bandwidth selection

In this section, we introduce a data-driven bandwidth selection procedure for the kernel estimator of the conditional distribution. In conformal inference, Liang et al. (2024a) has

considered selecting the score function to construct shortest prediction intervals. Here we instead focus on the localization parameter $h$ in our estimator. While extensive methods exist for bandwidth selection in kernel estimators, including rule-of-thumb approaches (Silverman, 1986) and cross-validation, these methods are primarily designed to improve estimation accuracy. As a more target-oriented approach, Guan (2023) selects the bandwidth $h$ by maximizing efficiency of prediction intervals on the training data. In the novelty detection problem, we instead consider maximizing the testing power, which best fits our goal.

Inspired by Zhang et al. (2022), the bandwidth which yields the largest number of discoveries is roughly the optimal one with the largest power with FDR control. Hence, the natural approach is running Algorithm 1 with different bandwidth $h$ in the kernel function $H$, recording the corresponding number of rejections and taking the bandwidth that leads to the largest number of rejections. However, directly running Algorithm 1 with the selected bandwidth can lead to inflated FDR due to post-selection effects. To cope with this issue, we need to investigate the structure of ALCP and try to preserve its validity.

Recall that the validity of ALCP $p_{\text{AL},j}$ hinges on the exchangeability of $\mathcal{D}_{c,j}$. To preserve this property while using a data-driven bandwidth, it is crucial that the bandwidth remains fixed given the unordered set of $\mathcal{D}_{c,j}$. Along this line, we propose a slight modification to the testing procedure when determining the bandwidth for the estimator $\widehat{F}_j$.

Suppose we have a set of candidate bandwidths $\Lambda$. For each $j \in \mathcal{U}$, we first treat it as a null sample and consider $\mathcal{D}_{c,j}$ and $\mathcal{D}_{u,j}$ as the new calibration and test datasets. Next, for each candidate $h \in \Lambda$, we compute ALCP set

$$\{p_{\text{AL},h,k}^{(j)}\}_{k \in \mathcal{U}_j} = \mathcal{A}(\mathcal{D}_{c,j}, \mathcal{D}_{u,j}, h).$$

Applying the BH procedure to these $p$-values yields the number of rejections

$$\gamma_{h,j} = |\text{BH}_\alpha(\{p_{\text{AL},h,k}^{(j)}\}_{k \in \mathcal{U}_j})|.$$

The bandwidth for $\widehat{F}_j$ in (4) is selected as the one that results in the largest number of rejections $h_j = \arg\max_{h \in \Lambda} \gamma_{h,j}$. Note that in Algorithm 1, permuting $\mathcal{D}_c$ does not affect the value of ALCP. Consequently, the number of rejections and optimal bandwidth for $\widehat{F}_j$ remain fixed given the unordered set of $\mathcal{D}_{u,j}$ in the above procedure.

Once the validity of ALCP is preserved, we can compute each $p_{\text{AL},j}^{\text{ada}}$ using its "optimal" bandwidth $h_j$ and proceed with the conditional calibration procedure as usual. We further take this bandwidth $h_j$ for computing auxiliary $p$-values $p_{\text{AL},k}^{(j),\text{ada}}$ for $k \in \mathcal{U}_j$. The complete procedure is summarized in Algorithm 2 in Appendix B. As previously illustrated, our procedure with data-driven bandwidths continues to ensure finite-sample FDR control.

**Remark 3.** *Notably, although the computational complexity is $O(|\Lambda| \cdot m^2)$, adopting an adaptive bandwidth does not substantially increase computational complexity. This is because the auxiliary $p$-values $p_{\text{AL},k}^{(j)}$ and kernel function $K(X_i, X_j; h_s)$ can be reused. Generally, the whole procedure runs up to 30s on a single core with $m$ on the order of hundreds. The selection procedure also has a clear structure for parallelization. When $m$ is very large, using parallelization can still control the runtime within several minutes.*

**Theorem 4.** *Suppose that the density ratio estimator $\widehat{r}$ is permutation invariant with respect to $\mathcal{D}_c \cup \mathcal{D}_u$. For any $j \in [m]$ such that $\mathcal{D}_{c,j}$ are exchangeable, the ALCP with data-driven bandwidths is valid in that*

$$\Pr(p_{\text{AL},j}^{\text{ada}} \leq t) \leq t, \quad 0 \leq t \leq 1.$$

*Furthermore, the rejection set $\mathcal{R}^{\text{ada}}$ output by Algorithm 2 ensures $\text{FDR} \leq \alpha$.*

## 4 Experiments and evaluation

In this section, we demonstrate the validity and efficiency of the proposed $p$-value and novelty detection procedure through simulated and real-data experiments, as detailed in Sections 4.1 and 4.2. Additional experiments are deferred to Appendix E. All experimental results are based on 200 replications.

## 4.1 Simulated experiments

We first validate the performance of our novelty detection method on simulated datasets. Here we consider a heterogeneous linear regression model, which is similar to the labeled scenario in Wu et al. (2025).

**Data generation**: The data vector is $Z = (X, W) = (X_1, \ldots, X_d, W) \in \mathbb{R}^{d+1}$ with $d = 10$. The model is $W = X_{-d}^\top \boldsymbol{\beta} + 5(1 + |X_d|)^{-1} \cdot \varepsilon$, with $X_1, \ldots, X_{d-1} \sim \mathrm{U}[-1, 1]$, $X_d \sim N(0, 1)$ and $\varepsilon \sim N(0, 1)$ independently. The coefficient vector is $\boldsymbol{\beta} = (0.5, -0.5, 0.5, -0.5, 0.5, 0, 0, 0, 0)^\top$.

The test data contains 20% novelties following the model $W = X_{-d}^\top \boldsymbol{\beta} + 5\left\{1 + (1 + |X_d|)^{-1}\right\} \cdot \xi$, where $\Pr(\xi = \pm 1) = 1/2$. We take the nominal level $\alpha = 0.2$, the weight parameter $\omega = 0.25$ and training and calibration sample sizes $|\mathcal{T}| = |\mathcal{C}| = 500$. Other details of the simulation setting can be found in Appendix B.

### 4.1.1 Comparison with baselines

First, we compare the performance of our method with existing novelty detection methods. We consider the following three methods:

- CP: the novelty detection method with marginal $p$-values in Bates et al. (2023) .

- LCP: the novelty detection method with localized $p$-values in Wu et al. (2025).

- ALCP: our proposed novelty detection method in Algorithm 2.

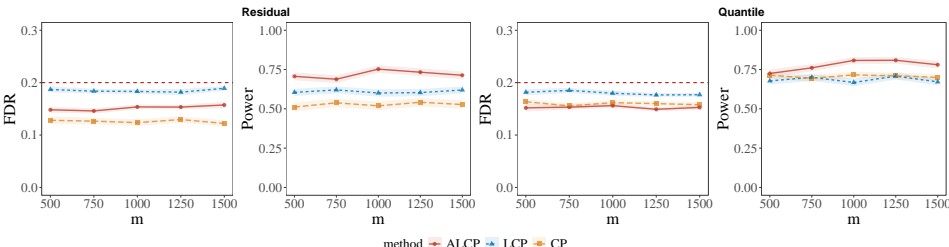

Figure 1: FDR and power of different methods under test sample sizes $m \in \{500, 750, 1000, 1250, 1500\}$ with residual (top row) and quantile (bottom row) scores.

**Results.** Results for comparing ALCP to benchmarks are summarized in Figure 1. As theoretically guaranteed, all three methods control the FDR below the nominal level. From the power perspective, ALCP consistently exhibits higher power compared to the other two methods. When comparing two score functions, ALCP maintains high power while the other methods show reduced performance with the residual score. This demonstrates that ALCP more effectively captures information from the conditional distribution, performing well even with a homogeneous score function.

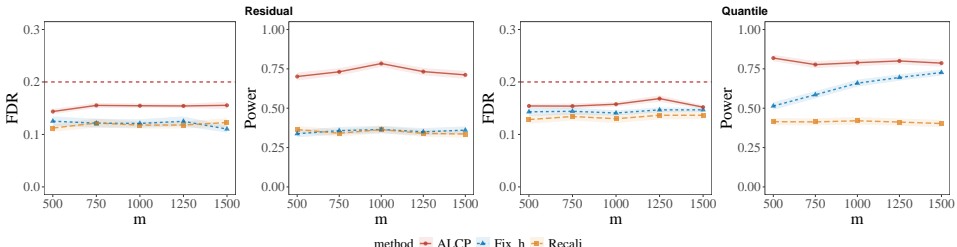

Figure 2: FDR and power of different algorithms under test sample sizes $m \in \{500, 750, 1000, 1250, 1500\}$ with residual (top row) and quantile (bottom row) scores.

### 4.1.2 ABLATION STUDIES

Here we show the contribution of each part in the construction of the ALCP method by comparing the performance of Algorithm 1, 2 and the method without utilizing the test data:

- Recali: replacing the $p$-values in Algorithm 1 with the recalibrated $p$-value in (3).
- Fix_h: Algorithm 1 with a fixed bandwidth.
- ALCP: Algorithm 2 with adaptive bandwidths.

**Results.** Results for ablation studies are summarized in Figure 2. While all three versions control the FDR below the nominal level, their power exhibits a stepwise increasing pattern. By integrating information from the test data, Fix_h achieves higher power than the simplified recalibration version with quantile score. Its power further increases with larger test sample sizes, which validates our theoretical conclusion on efficiency. The ALCP consistently demonstrates the highest power across both score functions and all sample sizes. This corresponds to our discussion that ALCP is more robust with a data-driven bandwidth compared with a fixed one.

### 4.2 REAL WORLD NOVELTY: U.S. HOUSE PRICE

**Dataset.** We further evaluate our method on the publicly available House Prices: Advanced Regression Techniques dataset from Kaggle (Kaggle, 2016). The main features include `SalePrice`, `GrLivArea` (above-ground living area), `TotalBsmtSF` (basement area), `OverallQual` (building quality), and `Neighborhood` (encoded as high-price or not). We treat `SalePrice` as the response variable $W \in \mathbb{R}$, and the remaining features as $X \in \mathbb{R}^d, d = 4$. The goal is to detect novelties regarding the conditional distribution of $W$ given $X$.

**Implementation details.** Similar to the novelty detection setting in Song et al. (2007), we calculate scores by training a prediction model on the full dataset and take outliers as the 10% of samples with the largest nonconformity scores from both the high-price and low-price groups. We take `Neighborhood` as the variable used for kernel weighting. The nominal FDR level $\alpha$ is set as 0.2.

Table 1: FDR and power for the House Sales dataset. Bold numbers represent the best results. The brackets contain the standard errors.

| Score | Value | Method | | |
| --- | --- | --- | --- | --- |
| | | ALCP | LCP | CP |
| Quan | FDR | 0.140(.006) | 0.153(.009) | 0.133(.007) |
| | Power | **0.642**(.010) | 0.520(.016) | 0.529(.015) |
| Res | FDR | 0.152(0.005) | 0.168(0.006) | 0.150(0.006) |
| | Power | **0.792**(0.006) | 0.615(0.009) | 0.565(0.009) |

**Results.** The FDR and power results for the U.S. house price dataset are reported in Table 1. All three methods successfully control the FDR below the nominal level, while ALCP detects the largest number of novelties among them. These findings demonstrate that ALCP is applicable in complex real data.

## 5 CONCLUDING REMARK

We conclude the paper with two remarks. Firstly, while both LCP and ALCP rely on kernel-based conditional distribution estimators to construct $p$-values, kernel methods do have limitations including boundary effects and inefficiency in high-dimensional settings. Exploring modern machine learning-based approaches for conditional distribution estimation could address these challenges and further enhance performance.

Secondly, the proposed novelty detection procedure with adaptive bandwidth can become computationally burdensome when the candidate set $\Lambda$ is large. To accommodate a broader range of tuning parameters, it is essential to develop more computationally efficient algorithms that do not rely on the conditional calibration step while still ensuring finite-sample FDR control. We leave these two issues as directions for future research.

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

## A  Related works

**Novelty detection** Our work is also related to novelty detection problems. Classical novelty detection methods include model-based approaches (Jobe and Pokojovy, 2015; Boukerche et al., 2020) and machine learning-based algorithms (Liu et al., 2008; Erfani et al., 2016; Hariri et al., 2019). Most existing works that consider conditional novelties fall into these two categories (Song et al., 2007; Hong and Hauskrecht, 2015). Recent studies have explored the use of conformal inference for novelty detection (Bates et al., 2023; Marandon et al., 2024; Liang et al., 2024b). While ensuring finite-sample FDR control, these works primarily focus on the marginal case.

**Localized conformal inference.** This work is closely related to Guan (2023) and Hore and Barber (2024), both of which emphasize the effect of localized regions to construct conformal prediction intervals that are valid conditional on the covariate. Guan (2023) proposed using a kernel-weighted empirical distribution for calibration but required an adjusted quantile level that is difficult to compute. Hore and Barber (2024) alleviates this by incorporating a random sampling step. In contrast, our work focuses on testing problems and emphasizes the construction of $p$-values rather than prediction intervals.

**Conformalized multiple testing.** Conformalized multiple testing has attracted significant attention in recent years and has been used for novelty detection (Bates et al., 2023; Liang et al., 2024b) and sample selection (Jin and Candès, 2023a;b; Wu et al., 2024). These works primarily use the conformal $p$-values for multiple testing while controlling the FDR. Bai and Jin (2024) further proposes to select the optimal score function that maximizes testing power. In a similar spirit, we focus on selecting optimal bandwidth to maximize the power in novelty detection problem. As an extension to conditional testing, Wu et al. (2025) introduces the localized conformal $p$-value and applies it to several conditional testing problems. The augmented localized conformal $p$-value proposed in this paper extends this work in a non-trivial way.

## B  Novelty detection algorithm with adaptive bandwidths

The novelty detection procedure with adaptive bandwidth provided in Section 3.2 can be summarized as follows.

---

**Algorithm 2** Novelty detection with data-driven bandwidths

---

**Input:** Calibration data $\mathcal{D}_c$ and test data $\mathcal{D}_u$ with index sets $\mathcal{C}$ and $\mathcal{U}$; FDR target level $\alpha \in (0,1)$; Kernel function $H(\cdot, \cdot)$ with bandwidth $h$; Score function $V$; Density ratio estimator $\widehat{r}$; Weight parameter $\omega$; Candidate bandwidth set $\Lambda$

1: **for** $j \in \mathcal{U}$ **do**
2:     **for** $h \in \Lambda$ **do**
3:         Compute ALCP set $\{p_{\text{AL},h,k}^{(j)}\}_{k \in \mathcal{U}_j} = \mathcal{A}(\mathcal{D}_{c,j}, \mathcal{D}_{u,j}, h)$;
4:         Obtain the rejection number of the BH procedure $\gamma_{h,j} = |\text{BH}_\alpha(\{p_{\text{AL},h,k}^{(j)}\}_{k \in \mathcal{U}_j})|$;
5:         Take $h_j = \arg\max_{h \in \Lambda} \gamma_{h,j}$.
6:     **end for**
7:     Compute ALCP $p_{\text{AL},j}^{\text{ada}}$ by taking the $j$-th value in $\mathcal{A}(\mathcal{D}_c, \mathcal{D}_u, h_j)$;
8:     Compute auxiliary $p$-values $\{p_{\text{AL},k}^{(j),\text{ada}}\}_{k \in \mathcal{U}_j} = \mathcal{A}(\mathcal{D}_{c,j}, \mathcal{D}_{u,j}, h_j)$;
9:     Compute $\mathcal{R}_j^{\text{ada}} = \text{BH}_\alpha(\{p_{\text{AL},k}^{(j),\text{ada}}\}_{k \in \mathcal{U}_j} \cup \{0\})$;
10: **end for**
11: Compute the final rejection set $\mathcal{R}^{\text{ada}}$ as in (8) with $p_{\text{AL},j}, p_{\text{AL},k}^{(j)}$ and $\mathcal{R}_j$ replaced by verions in steps 7-9.

**Output:** Detected novelty set $\mathcal{R}^{\text{ada}}$.

---

## C  Simulation details

Since our goal is to test for novelties regarding $W \mid X_d$, we only use $X_d$ for localization with true conditional dimension $d_{\mathrm{con}} = 1$. In all experiments we take the Gaussian kernel $H(x, x') = (2\pi h^2)^{-d_{\mathrm{con}}/2} \exp\{-\|x - x'\|_2^2/(2h^2)\}$ with $h = |\mathcal{C}|^{-1/(2(2+d_{\mathrm{con}}))}$, and the candidate bandwidths for ALCP is take as $\Lambda = \{10^{\lambda}|\mathcal{C}|^{-1/(2(2+d_{\mathrm{con}}))} : \lambda \in \{-1, -0.5, 0, 0.5, 1\}\}$. For the score function, we consider both the residual score $V(x, w) = |w - \widehat{\mu}(x)|$ with $\widehat{\mu}$ trained by the random forest algorithm and the quantile regression score $V(x, w) = \max\{w - \widehat{q}_{\alpha_{\mathrm{hi}}}(x), \widehat{q}_{\alpha_{\mathrm{lo}}}(x) - w\}$ with $\alpha_{\mathrm{lo}} = 0.1, \alpha_{\mathrm{hi}} = 0.9$ and $\widehat{q}$ trained by the quantile forest algorithm. The density ratio estimator $\widehat{r}$ is fitted by training a GLM model with $\{V_i\}_{i \in \mathcal{T}}$ and $\{V_j\}_{j \in \mathcal{C} \cup \mathcal{U}}$ as two classes. The probability output is taken as the function value of $\widehat{r}$.

For the U.S. house price dataset, we first train a randomForest and a quantileForest on the full dataset to obtain residual scores and quantile regression scores, which are then used to identify outliers. After excluding these outliers from the main dataset, we partition the remaining observations into three subsets: 20% are allocated to the null test set, 40% to the calibration set, and the remaining 40% to the training set. Among the defined outliers, we randomly select 60% to serve as outlier observations. Subsequently, the null test set is combined with the selected outliers to construct the final test set. The feature `Neighborhood` is encoded as a binary indicator of high-price versus low-price areas according to `SalePrice`: the low-price neighborhoods include `Edwards`, `OldTown` and `IDOTRR`, while the high-price neighborhoods include `Sawyer`, `Timber`, `NridgHt`, `CollgCr`, `NoRidge` and `StoneBr`. The bandwidths are chosen as $h = |\mathcal{C}|^{-1/(2(d_{\mathrm{con}}+2))}$ with $d_{\mathrm{con}} = 1$ for the LCP method, and $\Lambda = \{10^{-\lambda}|\mathcal{C}|^{-1/(2(2+d_{\mathrm{con}}))} : \lambda \in \{-1, -0.5, 0, 0.5, 1\}\}$ for the ALCP method.

## D  Proof of theorems

### D.1  Proof of Theorem 1

Since $\widehat{r}$ is permutation-invariant w.r.t. $\mathcal{D}_c \cup \mathcal{D}_u$ and therefore $\mathcal{D}_{c,j}$, we know the function $\widehat{F}_j(v, x)$ is also fixed if we permute the set $\mathcal{D}_{c,j}$. Combined with the assumption that $\mathcal{D}_{c,j}$ are exchangeable, we know $\{\widehat{F}_j(V_i, X_i)\}_{i \in \mathcal{C}_j}$ are also exchangeable since they are obtained by applying a function invariant to permutations on $\mathcal{D}_{c,j}$. The remaining prove follows exactly the same to the proof of finite sample coverage of conformal prediction intervals:

$$
\begin{aligned}
\mathrm{Pr}(p_{\mathrm{AL},j} \leq t) &= \mathrm{Pr}\left(\frac{\sum_{i \in \mathcal{C}} \mathbb{I}\{\widehat{F}_j(V_j, X_j) \leq \widehat{F}_j(V_i, X_i)\} + 1}{|\mathcal{C}| + 1} \leq t\right) \\
&= \mathrm{Pr}\left(\widehat{F}_j(V_j, X_j) \leq Q\left(t, \sum_{k \in \mathcal{C}_j} \frac{1}{|\mathcal{C}| + 1}\delta_{\widehat{F}_j(V_k, X_k)}\right)\right) \\
&= \frac{1}{|\mathcal{C}| + 1}\mathbb{E}\left[\sum_{k \in \mathcal{C}_j} \mathbb{I}\left\{\widehat{F}_j(V_k, X_k) \leq Q\left(t, \sum_{k \in \mathcal{C}_j} \frac{1}{|\mathcal{C}| + 1}\delta_{\widehat{F}_j(V_i, X_i)}\right)\right\}\right] \\
&= \frac{\lfloor(|\mathcal{C}| + 1)t\rfloor}{|\mathcal{C}| + 1} \leq t,
\end{aligned}
$$

where the third equality follows from the exchangeability. $\qquad\square$

### D.2  Proof of Theorem 2

The estimation $\widehat{F}_j(v, x)$ is

$$
\widehat{F}_j(v, x) = \frac{\sum_{i \in \mathcal{C}_j} H(X_i, x)\mathbb{I}\{V_i \leq v\} + \omega \sum_{k \in \mathcal{U}_j} H(X_k, x)\widehat{r}(V_k, X_k)\mathbb{I}\{V_k \leqslant v\}}{\sum_{i \in \mathcal{C}_j} H(X_i, x) + \omega \sum_{k \in \mathcal{U}_j} H(X_k, x)\widehat{r}(V_k, X_k)}.
$$

Denote $V_i \mid X_i \sim F_{V|X=X_i}(\cdot), i = 1, \cdots, n$, $V_k \mid X_k \sim F_{V'|X=X_k}(\cdot), k = n+1, \cdots, n+m$, then the difference is

$$|\widehat{F}_j(v,x) - F_j(v,x)| =$$

$$\frac{|\sum_{i \in \mathcal{C}_j} H(X_i,x) \left(\mathbb{I}\{V_i \le v\} - F_{V|X=x}(v)\right) + \omega \sum_{k \in \mathcal{U}_j} H(X_k,x)\widehat{r}(V_k,X_k) \left(\mathbb{I}\{V_k \le v\} - F_{V|X=x}(v)\right)|}{\sum_{i \in \mathcal{C}_j} H(X_i,x) + \omega \sum_{k \in \mathcal{U}_j} H(X_k,x)\widehat{r}(V_k,X_k)}$$

$$\le \frac{\sum_{i \in \mathcal{C}_j} H(X_i,x)|\mathbb{I}\{V_i \le v\} - F_{V|X=x}(v)| + \omega \sum_{k \in \mathcal{U}_j} H(X_k,x)\widehat{r}(V_k,X_k)|\mathbb{I}\{V_k \le v\} - F_{V|X=x}(v)|}{\sum_{i \in \mathcal{C}_j} H(X_i,x) + \omega \sum_{k \in \mathcal{U}_j} H(X_k,x)\widehat{r}(V_k,X_k)}.$$

Denote

$$A = \sum_{i \in \mathcal{C}_j} H(X_i,x)|\mathbb{I}\{V_i \le v\} - F_{V|X=x}(v)| + \omega \sum_{k \in \mathcal{U}_j} H(X_k,x)\widehat{r}(V_k,X_k)|\mathbb{I}\{V_k \le v\} - F_{V|X=x}(v)|,$$

$$A_1 = \sum_{i \in \mathcal{C}_j} H(X_i,x)|\mathbb{I}\{V_i \le v\} - F_{V|X=x}(v)|,$$

$$A_2 = \omega \sum_{k \in \mathcal{U}_j} H(X_k,x)r(V_k,X_k)|\mathbb{I}\{V_k \le v\} - F_{V|X=x}(v)|,$$

$$A_3 = \omega \sum_{k \in \mathcal{U}_j} H(X_k,x)(\hat{r}(V_k,X_k) - r(V_k,X_k))|\mathbb{I}\{V_k \le v\} - F_{V|X=x}(v)|,$$

$$B = \sum_{i \in \mathcal{C}_j} H(X_i,x) + \omega \sum_{k \in \mathcal{U}_j} H(X_k,x)\widehat{r}(V_k,X_k).$$

Then

$$|\widehat{F}_j(v,x) - F_{V|X=x}(v)| \le \frac{A}{B} = \frac{A_1 + A_2 + A_3}{B}.$$

In the following steps, $\mathbb{E}(A_1^2)$ and $\mathbb{E}(A_2^2)$ will be calculated to show their order with chebyshev's inequality. For simplicity, we only consider $(X_j, V_j)$ follows $\mathbb{H}_0$ since it only affect $H(X_j, x)\mathbb{I}\{V_j \le v\}$. For $A_1$, we have

$$\mathbb{E}\left[A_1^2\right] = (|\mathcal{C}| + 1)\mathbb{E}\left[H^2(X_i,x)\left(1\{V_i \le v\} - F_{V|X=x}(v)\right)^2\right]$$
$$+ |\mathcal{C}|(|\mathcal{C}| + 1)\mathbb{E}^2\left[H(X_i,x)\left(1\{V_i \le v\} - F_{V|X=x}(v)\right)\right].$$

Where $(X_i, V_i)$ is a data point in calibration set. For the first term, we have

$$\mathbb{E}[H^2(X_i,x)\left(1\{V_i \le v\} - F_{V|X=x}(v)\right)^2] \le \mathbb{E}\left[H^2(X_i,x)\right]$$
$$= \int_{\mathbb{R}^d} H^2(X_i,x) f(X_i)\, \mathrm{d}X_i$$
$$= \int_{\mathbb{R}^d} h^{-2d}K^2\left(\frac{\|X_i - x\|_2}{h}\right) f(X_i)\, \mathrm{d}X_i$$
$$\overset{(i)}{\le} \|f(X_i)\|_\infty \int_{\mathbb{R}^d} h^{-2d}K^2\left(\frac{\|X_i - x\|_2}{h}\right) \mathrm{d}X_i$$
$$\overset{(ii)}{=} \|f(X_i)\|_\infty \int_{\mathbb{R}^d} h^{-2d}K^2(\|t\|_2)\mathrm{d}(x + th)$$
$$\le \|f(X_i)\|_\infty \int_{\mathbb{R}^d} h^{-d}K^2(\|t\|_2)\mathrm{d}t$$
$$= \|f(X_i)\|_\infty h^{-d} \int_{\mathbb{R}^d} K^2(\|t\|_2)\mathrm{d}t. \tag{9}$$

The inequality (i) holds because $f(X_i) < \|f(X_i)\|_\infty$ and the equality (ii) replace $\frac{X_i-x}{h}$ with $t$. For the second term, following the same technique as in the first term we have

$$\mathbb{E}\left[H\left(X_i, x\right)\left(1\left\{V_i \leqslant v\right\} - F_{V|X=x}(v)\right)\right]$$

$$=\mathbb{E}[\mathbb{E}\left[H\left(X_i, x\right)\left(1\left\{V_i \leqslant v\right\} - F_{V|X=x}(v)\right)\right] \mid X_i]$$

$$=\mathbb{E}\left[H\left(X_i, x\right)\left(F_{V|X=X_i}(v) - F_{V|X=x}(v)\right)\right]$$

$$\leqslant \int_{\mathbb{R}^d} h^{-d} K\left(\frac{\|X_i - x\|_2}{h}\right) L \|X_i - x\|_2^\beta f(X_i)\mathrm{d}X_i$$

$$=L\|f(X_j)\|_\infty \int_{\mathbb{R}^d} h^{-d} K(\frac{\|X_i - x\|_2}{h}) \|X_i - x\|_2^\beta \, \mathrm{d}X_j$$

$$=L\|f(X_j)\|_\infty \int_{\mathbb{R}^d} K(\|t\|_2)^\beta h^\beta \mathrm{d}t$$

$$=L\|f(X_j)\|_\infty h^\beta \int_{\mathbb{R}^d} K(\|t\|_2) \|t\|_2^\beta \, \mathrm{d}t, \tag{10}$$

where $F_{V|X=x}(v)$ is the conditional CDF of $V$ evaluated at $V = v$ given $X = x$.

Combine (9) and (10), with chebyshev's inequality, for large enough $M$, we can get:

$$P\left(|A_1| \geqslant \sqrt{M\left[(n+1)\|f(X_i)\|_\infty h^{-d} \int_{\mathbb{R}^d} K^2(\|t\|_2)\mathrm{d}t + n(n+1)L\|f(X_j)\|_\infty h^\beta \int_{\mathbb{R}^d} K(\|t\|_2)\|t\|_2^\beta \, \mathrm{d}t\right]}\right) \leqslant \frac{1}{M}.$$

Then

$$A_1 = O_p(\sqrt{nh^{-d} + n^2 h^\beta}). \tag{11}$$

For $A_2$, we have

$$\mathbb{E}\left[A_2^2\right] = \omega^2(|\mathcal{U}|-1)\mathbb{E}\left[H^2\left(X_k, x\right) r\left(V_k, X_k\right)\left(\mathbb{I}\{V_k \leqslant v\} - F_{V|X=x}(v)\right)^2\right]$$
$$+ \omega^2(|\mathcal{U}|-1)|\mathcal{U}|\mathbb{E}^2\left[H\left(X_k, x\right) r\left(V_k, X_k\right)\left(\mathbb{I}\{V_k \leqslant v\} - F_{V|X=x}(v)\right)\right].$$

Where $(X_k, V_k)$ is a data point in test set. For the first term, we have

$$\mathbb{E}\left[H^2\left(X_k, x\right) r\left(V_k, X_k\right)\left(\mathbb{I}\{V_k \leqslant v\} - F_{V|X=x}(v)\right)^2\right] \leqslant \|r\left(V_k, X_k\right)\|_\infty^2 \mathbb{E}\left[H^2\left(X_k, x\right)\right]$$
$$\leqslant \|r\left(V_k, X_k\right)\|_\infty^2 \|f(X_k)\|_\infty h^{-d} \int_{\mathbb{R}^d} K^2(\|t\|_2)\mathrm{d}t. \tag{12}$$

For the second term,

$$\mathbb{E}\left[H\left(X_k, x\right) r\left(V_k, X_k\right)\left(\mathbb{I}\{V_k \leqslant v\} - F_{V|X=x}(v)\right)\right]$$

$$=\mathbb{E}\left[\mathbb{E}\left[H\left(X_k, x\right) r\left(V_k, X_k\right)\left(\mathbb{I}\{V_k \leqslant v\} - F_{V|X=x}(v)\right)\right] \mid X_k\right]$$

$$=\mathbb{E}\left[H\left(X_k, x\right) \mathbb{E}\left[r\left(V_k, X_k\right)\left(\mathbb{I}\{V_k \leqslant v\} - F_{V|X=x}(v)\right)\right] \mid X_k\right]$$

$$=\mathbb{E}\left[H\left(X_k, x\right)\left(F_{V|X=X_k}(v) - F_{V|X=x}(v)\right)\right]$$

$$\leqslant L\|f(X_k)\|_\infty h^\beta \int_{\mathbb{R}^d} K(\|t\|_2)\|t\|_2^\beta \, \mathrm{d}t. \tag{13}$$

Combine (12) and (13), follow the same procedure as $A_1$, we can get

$$A_2 = O_p(\omega\sqrt{mh^{-d} + m^2 h^\beta}). \tag{14}$$

The we find the order of $A_3$ by calculating $\mathbb{E}\left[|A_3|\right]$, that is

$$\mathbb{E}\left[|A_3|\right] = \omega(m-1)\mathbb{E}\left[\left|H(X_k, x)(\hat{r}(V_k, X_k) - r(V_k, X_k))(\mathbb{I}\{V_k \leqslant v\} - F_{V|X=x}(v))\right|\right]$$

$$\leqslant \omega(m-1)\mathbb{E}\left[|H(X_k, x)(\hat{r}(V_k, X_k) - r(V_k, X_k))|\right]$$

$$\leqslant \omega(m-1)\sqrt{\mathbb{E}\left[H^2\left(X_k, x\right)\right]\mathbb{E}\left[(\hat{r}\left(V_k, X_k\right) - r\left(V_k, X_k\right))^2\right]}$$

$$\leqslant \omega m h^{-d/2}\sqrt{\mathbb{E}\left[(\hat{r}\left(V_k, X_k\right) - r\left(V_k, X_k\right))^2\right]}.$$

So with Markov's inequality, for large enough $M$, we have

$$P\left(|A_3| \geqslant M\omega m h^{-d/2}\sqrt{\mathbb{E}\left[\left(\hat{r}\left(V_k, X_k\right) - r\left(V_k, X_k\right)\right)^2\right]}\right) \leqslant \frac{1}{M}.$$

Then

$$A_3 = O_p(\omega m h^{-d/2}\sqrt{\mathbb{E}\left[\left(\hat{r}\left(V_k, X_k\right) - r\left(V_k, X_k\right)\right)^2\right]}). \tag{15}$$

For $B$, we have :

$$\begin{aligned}
B &= \sum_{i \in \mathcal{C}_j} H(X_i, x) + \omega \sum_{k \in \mathcal{U}_j} H(X_k, x)\hat{r}(V_k, X_k) \\
&= \sum_{i \in \mathcal{C}_j} H(X_i, x) + \omega \sum_{k \in \mathcal{U}_j} H(X_k, x)r(V_k, X_k) \\
&\qquad + \omega \sum_{k \in \mathcal{U}_j} H(X_k, x)(\hat{r}(V_k, X_k) - r(V_k, X_k)). \tag{16}
\end{aligned}$$

In equality (16), item of the first term and the second term have bounded expectation and item of the third term is $o_p(1)$, so by LLN, we derive $B/(n + \omega m) \xrightarrow{p} \mathbb{E}\left[H\left(X_i, x\right)\right]$. For $\mathbb{E}\left[H\left(X_i, x\right)\right]$, we have

$$\begin{aligned}
\mathbb{E}\left[H\left(X_i, x\right)\right] &= \int_{\mathbb{R}^d} h^{-d} K(\frac{\|X_i - x\|_2}{h})f(X_i)\mathrm{d}X_i \\
&= \int_{\mathbb{R}^d} K(\|t\|_2)(f(x) + o(1))\mathrm{d}t \\
&\overset{(i)}{=} f(x) + o(1) \\
&\geqslant \frac{f(x)}{2}. \tag{17}
\end{aligned}$$

The equation (i) relies on the continuity of density function. Combine (11), (14), (15) and (17), we have

$$\begin{aligned}
\frac{A}{B} &= O_p\left(\frac{\sqrt{nh^{-d} + n^2 h^\beta} + \omega\sqrt{mh^{-d} + m^2 h^\beta} + \omega m h^{-d/2}\sqrt{\mathbb{E}\left[\left(\hat{r}\left(V_k, X_k\right) - r\left(V_k, X_k\right)\right)^2\right]}}{n + \omega m}\right) \\
&= O_p\left(\sqrt{h^\beta + \frac{1}{(n + \omega m)h^d}} + \frac{\omega m}{n + \omega m} h^{-d/2}\sqrt{\mathbb{E}\left[\left(\hat{r}\left(V_k, X_k\right) - r\left(V_k, X_k\right)\right)^2\right]}\right).
\end{aligned}$$

Then the proof is completed.

### D.3  PROOF OF THEOREM 3

Given the validity of ALCP, the remaining proof is similar to the general proof of conditional calibration (Fithian and Lei, 2022; Liang et al., 2024b). Define the conditional statistic $\Phi_j = ([(X_i, W_i)]_{i \in \mathcal{C}_j}, \{(X_k, W_k)\}_{k \in \mathcal{U}_j})$ where $[(X_i, W_i)]_{i \in \mathcal{C}_j}$ denotes the unordered set of $\{(X_i, W_i)\}_{i \in \mathcal{C}_j}$. We first prove that ALCP $p_{\mathrm{AL},j}$ is still super-uniform given $\Phi_j$ under the $j$-th null hypothesis. This is obviously by the fact that $\{(X_i, W_i)\}_{i \in \mathcal{C}_j}$ are i.i.d. under the $j$-th null hypothesis and conditional on the unordered set does not break the exchangeability.

Then we notice that the rejection set $\mathcal{R}_j$ is fixed given $\Phi_j$. Given the unordered set $[(X_i, W_i)]_{i \in \mathcal{C}_j}$ and each test point $(X_{n+k}, W_{n+k})$ with $k \neq j$, the value of $p_{\mathrm{AL},l}^{(j)}$ is fixed. So we have $\{p_{\mathrm{AL},l}^{(j)}\}_{l \neq j}$ fixed given $\Phi_j$. Since $\mathcal{R}_j$ is obtained by applying BH on $(p_{\mathrm{AL},1}^{(j)}, \ldots, p_{\mathrm{AL},j-1}^{(j)}, 0, p_{\mathrm{AL},j+1}^{(j)}, \ldots, p_{\mathrm{AL},m}^{(j)})$, it is also fixed given $\Phi_j$.

Now we can analyse the FDR term-wisely. For each $j \in \mathcal{I}$, let $\gamma_{j0}^*$ be the number of rejections of the algorithm if we replace $p_{\mathrm{AL},j}$ with 0. We have

$$\mathbb{E}\left(\frac{\mathbb{I}\{j \in \mathcal{R}\}}{|\mathcal{R}| \vee 1}\right) = \mathbb{E}\left(\frac{\mathbb{I}\{p_{\mathrm{AL},j} \le \alpha|\mathcal{R}_j|/m\} \cdot \mathbb{I}\{\zeta_j \le \gamma^*/|\mathcal{R}_j|\}}{|\mathcal{R}| \vee 1}\right)$$

$$= \mathbb{E}\left(\frac{\mathbb{I}\{p_{\mathrm{AL},j} \le \alpha|\mathcal{R}_j|/m\} \cdot \mathbb{I}\{\zeta_j \le \gamma_{j0}^*/|\mathcal{R}_j|\}}{\gamma_{j0}^*}\right)$$

$$= \mathbb{E}\left(\frac{\mathbb{I}\{p_{\mathrm{AL},j} \le \alpha|\mathcal{R}_j|/m\}}{|\mathcal{R}_j|}\right)$$

$$= \mathbb{E}\left\{\mathbb{E}\left(\frac{\mathbb{I}\{p_{\mathrm{AL},j} \le \alpha|\mathcal{R}_j|/m\}}{|\mathcal{R}_j|} \mid \Phi_j\right)\right\}$$

$$\stackrel{(i)}{=} \mathbb{E}\left\{\mathbb{E}\left(\mathbb{I}\{p_{\mathrm{AL},j} \le \alpha|\mathcal{R}_j|/m\} \mid \Phi_j\right) \frac{1}{|\mathcal{R}_j|}\right\}$$

$$\stackrel{(ii)}{=} \frac{\alpha}{m}.$$

Here $(i)$ and $(ii)$ follows from conclusions that $p_{\mathrm{AL},j}$ is valid given $\Phi_j$ and $\mathcal{R}_j$ is fixed given $\Phi_j$. Taking summation over $j \in \mathcal{U}$ leads to the FDR control conclusion. $\square$

### D.4 Proof of Theorem 4

By Algorithm 2, the optimal bandwidth $h_j$ is measureable with respect to $\Phi_j$. This indicates that $p_{\mathrm{AL},j}^{\mathrm{ada}}$ with bandwidth $h_j$ is still super-uniform conditional on $\Phi_j$ under the assumption that $\mathcal{D}_{u,j}$ are exchangeable. Since the following conditional calibration procedure is exactly the same and each auxiliary $p$-value vector $(p_{\mathrm{AL},1}^{(j),\mathrm{ada}}, \ldots, p_{\mathrm{AL},j-1}^{(j),\mathrm{ada}}, 0, p_{\mathrm{AL},j+1}^{(j),\mathrm{ada}}, \ldots, p_{\mathrm{AL},m}^{(j),\mathrm{ada}})$ is computed with $h_j$, the rejection set $\mathcal{R}_j^{\mathrm{ada}}$ is still fixed given $\Phi_j$. Therefore, the remaining proof follows exactly the same to the proof of Theorem 3. $\square$

## E  Additional experiments

In this section we provide more experimental results to further illustrate the performance of different methods considered in this paper. Section E-E.8 contain additional simulation results. All simulations are performed on the same setting to that in Section 4.1.1- 4.1.2 with different varying parameters or algorithms. If not otherwise stated, the parameters will be the same to those in the main text. Section E.11 contains an additional real-data experiment on a bike-sharing dataset.

### E.1  Effect of bandwidth selection

To further highlight the effect of using adaptive bandwidth, we provide how the selected $h_j$ distributes when the test sample size $m$ and the conditional variance $\sigma_W$ of $W \mid X$ varies. The labeled sample sizes are fixed at $|\mathcal{D}_t| = |\mathcal{D}_c| = 100$. We fix $\sigma_W = 0.1$ when $m$ varies and $m = 200$ when $\sigma_W$ varies.

From Figure 3, we observe clear trends in how the optimal bandwidth varies with $\sigma_W$ and $m$. As the variance $\sigma_W$ decreases or the sample size $m$ increases, the optimal bandwidth becomes smaller; conversely, larger $\sigma_W$ or smaller $m$ lead to a larger optimal bandwidth. These empirical patterns are consistent with the theoretical expression for the optimal bandwidth in kernel estimation.

### E.2  Estimation of $\hat{r}$

We evaluate the performance of our methods with the density ratio function $\hat{r}$ estimated using the SVM algorithm, in contrast to the GLM algorithm used in the main text. All

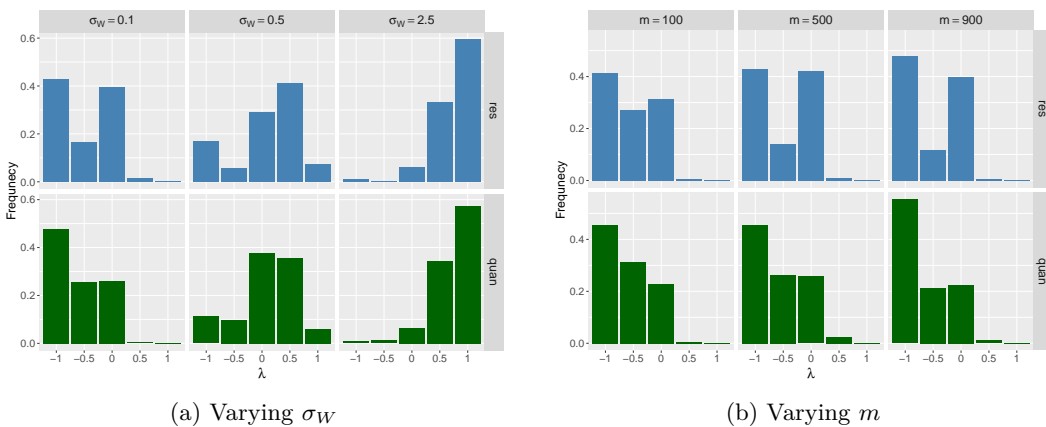

(a) Varying $\sigma_W$                       (b) Varying $m$

Figure 3: Distribution of optimal bandwidth $h_j$ for each test sample with varying $\sigma$ and $m$. The bandwidth candidate set is $\Lambda = \{10^{\lambda}|\mathcal{C}|^{-1/(2(2+d_{\text{con}}))} : \lambda \in \{-1, -0.5, 0, 0.5, 1\}\}$.

other settings remain identical to those in the main text, with a fixed test sample size of $m = 1000$.

Table 2: FDR and power results with $\widehat{r}$ estimated by two kinds of algorithms. Bold numbers represent the best results.

| Algorithm | Value | Residual score | | | Quantile score | | |
|---|---|---|---|---|---|---|---|
| | | ALCP | Fix_h | Recali | ALCP | Fix_h | Recali |
| GLM | FDR | 0.153 | 0.113 | 0.108 | 0.153 | 0.134 | 0.119 |
| | Power | **0.681** | 0.344 | 0.333 | **0.802** | 0.617 | 0.384 |
| SVM | FDR | 0.106 | 0.019 | 0.112 | 0.154 | 0.118 | 0.126 |
| | Power | **0.490** | 0.012 | 0.312 | **0.839** | 0.409 | 0.413 |

Results in Table 2 show that SVM algorithm could lead to a worse estimator than the GLM for the residual score since ALCP and Fix_h both have power reduced with this algorithm. Although Fix_h is severely influenced with the residual score, ALCP mitigates this issue with an adaptive bandwidth. For the quantile score, ALCP shows quite similar power across two algorithms, while Fix_h is still negatively affected. For both scores, ALCP is more robust compared with Fix_h, showing the advantage of using adaptive bandwidths.

### E.3  PARAMTER $\omega$

We investigate the influence of $\omega$ in our proposed methods by showing performance under different values of $\omega \in \{0.25, 0.5, 0.75\}$. The rest settings are fixed with $m = 1000$.

Table 3: FDR and power results under different values of $\omega$. Bold numbers represent the best results.

| $\omega$ | Value | Residual score | | | Quantile score | | |
|---|---|---|---|---|---|---|---|
| | | ALCP | Fix_h | Recali | ALCP | Fix_h | Recali |
| 0.25 | FDR | 0.157 | 0.120 | 0.118 | 0.154 | 0.140 | 0.125 |
| | Power | **0.753** | 0.339 | 0.349 | **0.791** | 0.648 | 0.397 |
| 0.5 | FDR | 0.154 | 0.134 | 0.116 | 0.154 | 0.148 | 0.140 |
| | Power | **0.745** | 0.375 | 0.335 | **0.752** | 0.749 | 0.384 |
| 0.75 | FDR | 0.151 | 0.139 | 0.117 | 0.160 | 0.153 | 0.138 |
| | Power | **0.744** | 0.432 | 0.317 | 0.783 | **0.806** | 0.440 |

From Table 3 we can see the value of $\omega$ only slightly influence the performance of our methods. Among all three values, ALCP still achieves the highest power in most cases except for $\omega = 0.75$ combined with the quantile score. In this case, the power of ALCP and Fix_h are quite close.

### E.4 RELATIONSHIP BETWEEN $\widehat{r}_1$ AND $w$

In Section 2.2, we briefly discussed the choice of $w$, noting that a larger $w$ should be preferred when the estimator $\widehat{r}_1$ is believed to be accurate. Here, we empirically validate this intuition using simulated evidence. Under the same setting as in Section 4.1.2 with $m = 500$, we compare ALCP with a variant that uses an oracle $r$. Because obtaining the exact distribution of machine-learningbased scores is infeasible, we approximate the oracle $r$ by training it on an extensive leave-out dataset whose size is ten times larger than that used in the main experiments. We then record and compare the power curves of the two methods across different values of $w$.

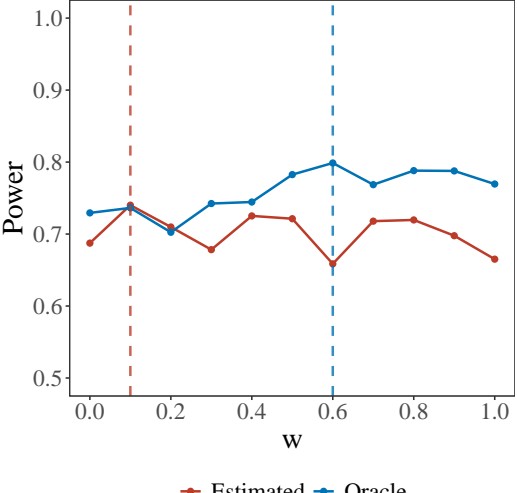

Figure 4: Power comparison of ALCP using the estimated density ratio $r$ and its oracle counterpart. The dotted lines highlight the highest power attained by each method.

Figure 4 shows that ALCP achieves relatively higher power when equipped with the oracle density ratio function compared to the estimated one. We also observe that the optimal choice of $w$ is around 0.6 for the oracle version, substantially larger than the optimal value of 0.1 obtained with the estimated version. This pattern provides empirical support for our proposed rule for selecting the parameter $w$.

### E.5 NOVELTY RATIO

We compare our ALCP method both with LCP, CP and our two simplified versions under different novelty ratios. We consider ratio in the range $\{0.1, 0.15, 0.2\}$ and keep the remaining settings as fixed with $m = 1000$.

Tables 4-5 contains results under different novelty ratios in the test data. As the ratio increases, the signal becomes stronger and all methods have their power improved. Except for this pattern, the relative performance in all comparisons remain the same to simulation results in the main text.

Table 4: FDR and power results under different novelty ratios for ALCP, LCP and CP. Bold numbers represent the best results.

| Ratio | Value | Residual score | | | Quantile score | | |
|---|---|---|---|---|---|---|---|
| | | ALCP | LCP | CP | ALCP | LCP | CP |
| 0.1 | FDR | 0.153 | 0.148 | 0.108 | 0.153 | 0.200 | 0.119 |
| | Power | **0.116** | 0.046 | 0.069 | **0.255** | 0.060 | 0.136 |
| 0.15 | FDR | 0.146 | 0.206 | 0.095 | 0.145 | 0.193 | 0.143 |
| | Power | **0.467** | 0.341 | 0.261 | **0.607** | 0.418 | 0.432 |
| 0.2 | FDR | 0.155 | 0.193 | 0.131 | 0.158 | 0.179 | 0.163 |
| | Power | **0.755** | 0.620 | 0.567 | **0.839** | 0.702 | 0.727 |

Table 5: FDR and power results under different novelty ratios for ALCP, Fix_h and Recali. Bold numbers represent the best results.

| Ratio | Value | Residual score | | | Quantile score | | |
|---|---|---|---|---|---|---|---|
| | | ALCP | Fix_h | Recali | ALCP | Fix_h | Recali |
| 0.1 | FDR | 0.080 | 0.039 | 0.038 | 0.098 | 0.047 | 0.057 |
| | Power | **0.119** | 0.035 | 0.026 | **0.217** | 0.093 | 0.045 |
| 0.15 | FDR | 0.142 | 0.082 | 0.087 | 0.155 | 0.117 | 0.106 |
| | Power | **0.477** | 0.140 | 0.135 | **0.594** | 0.351 | 0.173 |
| 0.2 | FDR | 0.142 | 0.112 | 0.126 | 0.153 | 0.137 | 0.134 |
| | Power | **0.668** | 0.346 | 0.329 | **0.799** | 0.611 | 0.399 |

### E.6 COMPARISON WITH ADADETECT

We compare our proposed ALCP method with AdaDetect under the same setting as Section 4.1.1. In this experiment, we focus on the quantile scores, and both methods successfully control the FDR at the target level. $\alpha$.

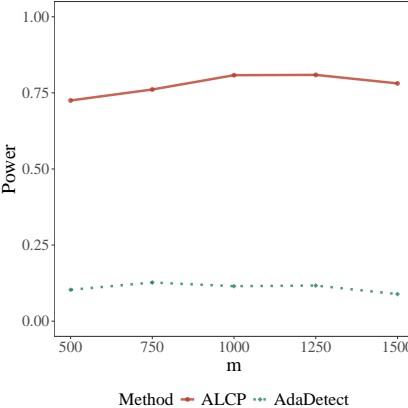

Figure 5: Power of ALCP and AdaDetect under different test sample sizes $m \in \{500, 750, 1000, 1250, 1500\}$.

Figure 5 display the power of both methods under different test sample sizes. Since AdaDetect is designed for marginal novelty detection and does not incorporate conditional distributional information, its statistical power is substantially lower than that of ALCP.

### E.7 TRAINING SAMPLE SIZE $|\mathcal{D}_t|$

We compare our ALCP method with LCP, CP, and AdaDetect under varying training sample sizes $|\mathcal{D}_t| \in \{500, 750, 1000, 1250, 1500\}$, while fixing the test sample size at $m = 1000$ and

keeping the other settings unchanged. Here we focus on the quantile scores, and all methods successfully control the FDR at the target level.

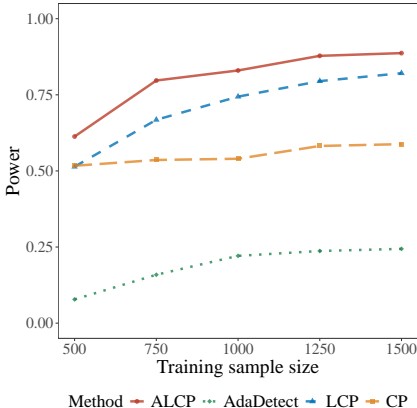

Figure 6: Power of different methods under varying training sample sizes $|\mathcal{D}_t| \in$ $500, 750, 1000, 1250, 1500$.

Figure 6 shows that as the training sample size increases, all methods derive higher power since the prediction model becomes more accurate. In addition, ALCP consistently achieves the highest power across different training sample size.

### E.8   Calibration sample size $|\mathcal{D}_c|$

We compare our ALCP method with LCP and CP under varying calibration sample sizes $n \in \{600, 800, 1000, 1200\}$, while fixing the test sample size at $m = 1000$ and keeping the other settings unchanged. Here we focus on the quantile scores, and all methods successfully control the FDR at the target level.

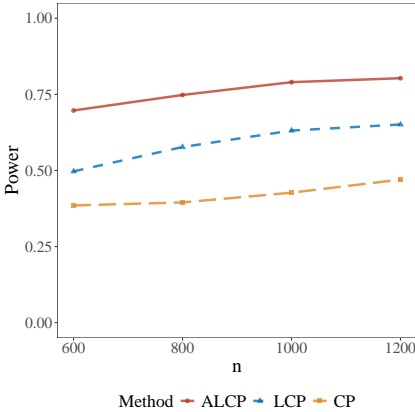

Figure 7: Power of different methods under varying calibration sample sizes $n \in$ $\{600, 800, 1000, 1200\}$.

Figure 7 shows that as the training sample size increases, power of all methods increase. The other conclusion remain the same to simulation results in the main text.

### E.9 INFLUENCE OF COVARIATE SHIFT

In Theorem 1, we require the marginal distributions satisfy $P_X = Q_X$ to ensure the ALCP is valid in finite sample. Here we illustrate ALCP is robust against certain degree of covariate shift. We take $m = 1000$ and consider both mean and variance shifts on the conditional variable $X_d$. That is, instead of $N(0,1)$, we have $X_d \sim N(\mu, 1)$ or $X_d \sim N(0, \sigma_X^2)$ for the test data. The distribution of clean labeled data is still fixed as $X_d \sim N(0,1)$. The score function is taken as the quantile score.

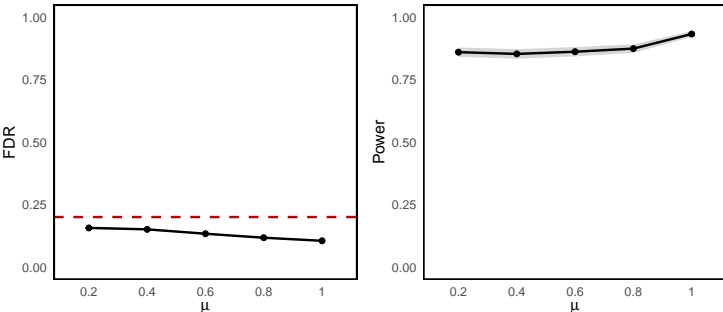

Figure 8: FDR and power of ALCP when there is mean shift in the test covaraites.

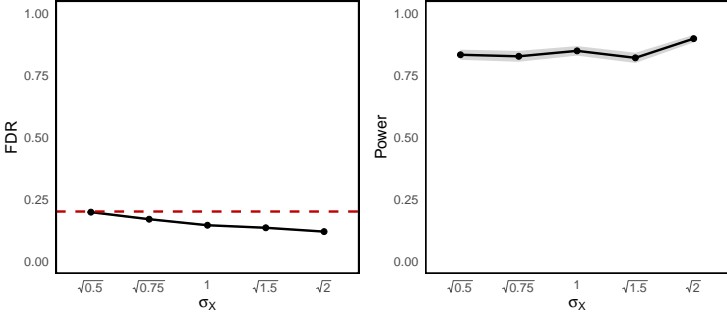

Figure 9: FDR and power of ALCP when there is variance shift in the test covaraites.

Figures 8-9 show that even the condition $P_X = Q_X$ does not hold, the ALCP procedure can still control the FDR within a certain range of covariate shift. It can be observed from the figures that mean shift and enlarged variance do not result in FDR increase. If the variance is reduced, the FDR will tend to inflate the nominal level. However, it is still controlled around the nominal level $\alpha$ even with a halved variance.

### E.10 CATEGORICAL RESPONSES

Since we focus on regression settings in the main text, here we provide an example where $W = 0, 1$ is categorical. The data is generated as follows:

- $X \sim U[0,1]$
- For inliers, $\Pr(W = 0 \mid X = x) = \frac{1}{1+\exp\{-10(x-0.5)\}}$
- For outliers, $\Pr(W = 0 \mid X = x) = 1 - \frac{1}{1+\exp\{-10(x-0.5)\}}$

Therefore, there is conditional distribution drift between the inliers and outliers. We take $|\mathcal{T}| = |\mathcal{C}| = 500$ with varying $m$. The score function is taken as the probability output of a

random-forest model trained on $\mathcal{D}_t$. The bandwidth for RLCP is taken as $h = 0.5$, and the candidate set for ALCP is taken as $\Lambda = \{0.25, 0.5, 0.75, 1\}$. All the remaining parameters are the same to experiments in the main text.

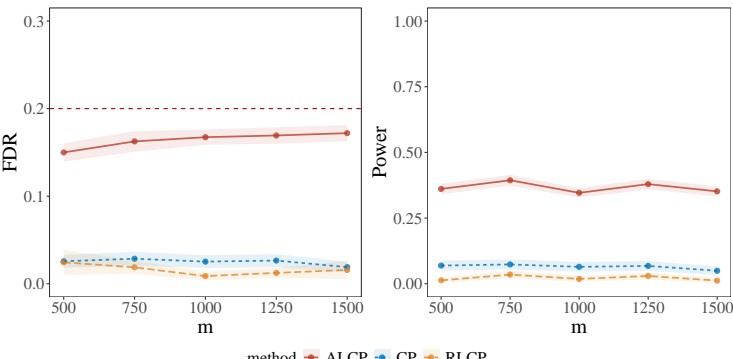

Figure 10: FDR and power of different methods under the classification scenario.

Figure 10 exhibits the same qualitative pattern as Figure 1 in the main text. The key difference is that all methods show reduced power in this setting. This degradation is primarily due to the discrete nature of $W$. Once $W$ becomes categorical, its conditional distribution given $X$ contains only two point masses, which substantially limits the amount of exploitable information in the detection procedure. Consequently, even when the inlier and outlier distributions differ significantly, the power is still below 0.5. In such scenarios, employing a well-calibrated predictive model with accurate probability estimates can be particularly beneficial.

### E.11 Temporal data: Bike sharing

**Dataset.** We use Bike Sharing dataset (Fanaee-T, 2013) to show the performance of our method on the temporal scenario, which has $17,389$ individuals with 13 features. Here we consider the hour in a day as the time feature $t$. The variable $W$ of interest is the number of total rental bikes. The goal is to detect novelties regarding the conditional distribution of $W$ given the time feature.

**Implementation details.** For each replicate, we randomly sample three parts from the whole dataset: $|\mathcal{T}| = 500$ training data, $|\mathcal{C}| = 500$ calibration data and $m = 1000$ test data. The synthetic novelties are constructed as

$$\tilde{W}_i = W_i + \varepsilon_i, \ \varepsilon_i = 0.5 \cdot \text{Quantile}_{0.8}(W \mid t = t_i) + \bar{W},$$

where $\bar{W}$ is the overall mean value of $W$ in the whole dataset. We take $t$ as the variable used for kernel weighting. The bandwidths are chosen as $h = 2$ for the LCP method and $\Lambda = \{1, 2, 3, 4, 5\}$ for the ALCP method.

Table 6: FDR and power for the Bike Sharing dataset with synthetic novelties. Bold numbers represent the best results. The brackets contain the standard errors.

| Score | Value | Method | | |
| --- | --- | --- | --- | --- |
| | | ALCP | LCP | CP |
| Res | FDR | 0.158(.006) | 0.238(.005) | 0.145(.007) |
| | Power | **0.573**(.021) | 0.393(.012) | 0.258(.008) |
| Quan | FDR | 0.165(.009) | 0.209(.005) | 0.149(.007) |
| | Power | **0.502**(.020) | 0.388(.012) | 0.383(.018) |

**Results.** The empirical FDR and power for Bike Sharing dataset are reported in Table 6. Notably, the dataset exhibits some temporal dependency, as indicated by a slightly

inflated FDR for the LCP method. In contrast, ALCP maintains FDR control below the nominal level $\alpha$ even under weak dependency and achieves the highest power across both score functions.

