# OpenReview forum: "Novelty Detection with Augmented Localized Conformal $p$-values"
_ICLR.cc/2026/Conference — Submitted to ICLR 2026_

### Official Review · Reviewer_UVEZ · 2025-10-25

**Soundness:** 2
**Presentation:** 3
**Contribution:** 2
**Rating:** 4
**Confidence:** 3

**Summary:**

The paper proposes ALCP for conditional novelty detection. It performs localized conformal calibration with kernel weights and augments the calibration using test data through density-ratio reweighting to handle shift from novel points, then applies a recalibration step that yields finite sample valid p-values and pairs with Benjamini–Hochberg to control the FDR. It also introduces a data driven bandwidth rule with a small correction to keep validity. On synthetic data and a real housing dataset, ALCP shows higher detection power at controlled FDR than localized and marginal conformal methods.

**Strengths:**

The paper introduces ALCP for conditional novelty detection, combining localized conformal calibration with test-set augmentation via density-ratio reweighting to handle shift from novel points; it provides finite-sample valid p-values and supports FDR control with Benjamini–Hochberg, includes a data-driven bandwidth rule with a correction that preserves validity, and shows higher detection power at controlled FDR on synthetic data and a real housing dataset.

**Weaknesses:**

- The contribution feels incremental. It extends localized conformal prediction with test-set augmentation and a data-driven bandwidth, and the route to FDR control seems to rely on existing results.
- The paper does not clearly explain how it differs from prior conformal methods or which guarantees are new, nor does it clearly lay out the challenges it aims to solve or rigorously reason through their theoretical and practical implications, so the proposed fixes to LCP read as a modest mix of existing ideas without a convincing impact.
- Bandwidth selection requires recomputing auxiliary p-values and running BH across many bandwidths, which may be impractical at scale or with many hyperparameters.
- There is little analysis of how kernel choice, density-ratio estimation, and randomization affect power; the bandwidth study does not report selected values or their variability. key baselines (standard anomaly detectors and closely related conformal methods) are missing, making the incremental benefit of the proposed score hard to judge.

**Questions:**

See Weaknesses.

---

> ### Author Response · Authors · 2025-11-21
>
> Thank you for your careful reading and thoughtful comments on our work. We would like to respond to weaknesses you raised pointwisely. All modifications we mentioned in the response are updated in the revised paper (it is available by clicking the pdf icon at the top of the website).
>
>
> 1–2. We would like to offer some clarifications regarding our contributions and how our work differs from the existing literature. Our work originates from the LCP method introduced in Wu et al. (2025). We adopt a similar problem setting but a different primary motivation, which is to address specific limitations inherent in the LCP approach. As discussed in the introduction, these limitations can lead to significant drawbacks in certain practical scenarios. To overcome these issues, we introduce a set of alternative techniques that enhance robustness and efficiency, leading to a more reliable ALCP method for conditional anomaly detection.
>
> First, we incorporate information from the test data to improve the efficiency of constructed p-values. This is achieved by using augmented kernel estimators, followed by a recalibration step. We also adjust for the distribution by adding density ratio terms. The recalibration step ensures finite-sample validity of the resulting p-values.
>
> To ensure finite-sample validity, the LCP method introduces a random sampling step for each test point. However, this introduces external randomness into the testing procedure, potentially compromising its stability. In contrast, our recalibration technique avoids any additional randomization and yields deterministic p-values.
>
> We also propose a novel bandwidth selection strategy aimed at optimizing selection power. By approximating the objective function, our method identifies bandwidths that enhance testing performance while still maintaining finite-sample FDR control.
>
> While the idea of improving an existing method using additional techniques may appear straightforward, the specific enhancements we introduce are far from trivial modifications. Integrating these components while preserving finite-sample theoretical guarantees requires careful methodological design to ensure that they work cohesively. This is precisely the main challenge we address. Hope this solves your concern!
>
> 3. Thank you for raising this issue. The computational complexity of a whole procedure with bandwidth selection is $O(|\Lambda| \cdot m^2)$. As discussed in Remark 3, the practical computational cost can be substantially reduced by reusing kernel evaluations and a base estimator, since a kernel estimator can be efficiently updated when only a single training point is modified. Moreover, the selection procedure has a natural structure for parallelization. When the bandwidth range is large, parallelization can reduce the runtime by a significant factor. For the problem considered in our paper, we also note that selecting bandwidths up to an order of 10 is typically sufficient, so the size of $\Lambda$ remains small in common applications. In the revised paper, we have also added more discussion on computation in Remark 3, Line 360.
>
> 4. We would like to clarify that potential baselines of novelty detection with finite-sample FDR control guarantee are limited (CP in Bates et al. (2023), AdaDetect in Marandon et al. (2024) and Wu et al. (2025)) and all are considered in our experiments. Per the advice of all reviewers, we have added extra experiments to further show the incremental benefit of ALCP. This includes the aspects you raised. We kindly refer you to Section E.1, E.4, E.9 and E.10 for new experiment results.

---

### Official Review · Reviewer_eRMP · 2025-10-30

**Soundness:** 3
**Presentation:** 3
**Contribution:** 2
**Rating:** 4
**Confidence:** 4

**Summary:**

This paper studies the conformal outlier detection for the conditional distribution shift. The authors design a framework to construct p-values based on kernel estimation that are sensitive to conditional shift, and leverage a randomization technique to preserve finite-sample validity of the p-values. The p-values are then coupled with a conditional calibration strategy to ensure valid FDR control. They also present a strategy to adaptively choose the bandwidth in kernel estimation while preserving the validity of p-values and FDR control. The power improvement of the proposed method is demonstrated in simulations and a real-world case study of housing price.

**Strengths:**

1. The paper is clearly written and easy to follow in general.
2. The use of various techniques are valid and sound. Although none of them seems completely new, a combination of them leads to a solution to a somewhat useful problem.

**Weaknesses:**

1. The conditional testing problem seems to be at odds with the FDR control, and the statement of conditions is a bit unclear (see my questions).
2. The scale of numerical evaluations is a bit limited.
3. The guarantees are still marginal FDR, which is not very new.
4. The theoretical understanding of power boost can be deepened.

**Questions:**

1. While the authors propose to test the null hypothesis $P_{Y|X}=Q_{Y|X}$, the FDR control is still with regard to a marginal exchangeability condition $P_{X,Y}=Q_{X,Y}$. Is my understanding correct? If yes, then the motivation and claim of addressing conditional shift may be insufficiently supported. In particular, it is possible that the null hypothesis $P_{Y|X}=Q_{Y|X}$ holds, but due to covariate shift the marginal null hypothesis $P_{X,Y}=Q_{X,Y}$ is not true and can be rejected, leading to inflated type-I error.
2. Following the above question, the simulations should also consider settings with covariate shift to show the robustness of the methods, even if the theory can only be limited to marginal test.
3. Are there theoretical results, such as asymptotic analysis, that can illustrate why your method can improve power?

---

> ### Author Response · Authors · 2025-11-21
>
> Thank you for your review and helpful comments on our paper. For all parts of your review (except for Strength), we would like to make some clarification and explanation pointwisely. All modifications we mentioned in the response are updated in the revised paper (it is available by clicking the pdf icon at the top of the website).
>
> ### Summary
>
> First, we would like to clarify the description of “leveraging a randomization technique to preserve finite-sample validity of the p-values.” In fact, our method does not use any randomization. On the contrary, removing the exogenous randomness in the LCP method (Wu et al., 2025) is part of our contribution. This point is also explicitly stated in the methodological derivation at Line 180 in Section 2.2. Hope it is clear now!
>
> ### Weaknesses
>
> 1. We would like to clarify about the problem we consider. While conditional testing is a collection of many different problems, conditional outlier detection is one of them with a natural multiple testing structure. Just like when testing for marginal outliers, the FDR is a commonly-used criterion from a statistical perspective. For some other conditional testing problems, different criteria including the FWER or type I error could be preferred (Wu et al., 2025), which totally depends on the specific problem.
>
> 2. Thank you for your advice. Based on the comments of all reviewers, we have added additional experiments to show the performance of our method more comprehensively. We kindly refer you to Section E.1, E.4, E.9 and E.10 in the revised paper for detailed experiment results.
>
> 3. Thank you for raising this issue. Following the original motivation of our work, we are considering a novelty detection problem. As practitioners typically care about the error rate over the entire test dataset, and providing a marginal control guarantee could be more suitable even when the goal is to detect conditional novelties. The word “conditional” in conditional novelty detection is more about the property of novelties (that is, whether the relationship between $W$ and $X$ has changed), rather than the type of error rate being controlled. Therefore, what we do on the conditional perspective is to capture more information of the conditional distribution by proposing new statistics, which we believe provides more practical value. This also is related to the issue of covariate shift, which we further discuss in our response to Question 1.
>
> 4. Presenting the power improvement in a circumstantial manner is mainly driven by the nature of the problem we consider. In conformalized multiple testing, power analysis is often given by proving the asymptotic power, which is a limit value depending on the score function and data distribution. In our method, we take the conditional CDF estimator as the second step score. Although the form of estimator in ALCP is different from that in LCP, their limit is the same (which is $F_{V \mid X = x}(v, x)$). Therefore, the asymptotic power will also be the same. While explicitly characterizing how the improved convergence rate translates into power gains is quite challenging, we view this as an interesting direction for future work. Hope this is acceptable to you!
>
> ### Questions
>
> 1–2. Thank you for your question. You are correct on the point that finite-sample FDR control relies on the assumption $P_X = Q_X$. The assumption is mainly introduced to provide a finite-sample valid theoretical result, which is important in conformal prediction literature. For cases when $P_X \neq Q_X$, our proposed ALCP p-value is still asymptotically valid. However, since asymptotically valid p-values cannot induce asymptotic FDR control, we decide not to include this less informative result in the paper. We have also performed additional experiments to show how covariate shift affects the ALCP method. In summary, the ALCP method can still control the FDR when there is a certain degree of covariate shift. Please see Section E.9 (Line 1136), Figures 8–9 for detailed results.
>
> 3. Please see our response to Weakness 4 for a unified discussion of this point.

---

### Official Review · Reviewer_1Phq · 2025-10-30

**Soundness:** 2
**Presentation:** 3
**Contribution:** 2
**Rating:** 6
**Confidence:** 4

**Summary:**

This paper addresses the problem of novelty detection, where the goal is to test whether a test point was drawn from an inlier distribution $P$ or from a different distribution $Q \neq P$. Specifically, the authors consider conditional novelty detection, where the inlier distribution takes the form $P=P_{W|X} \cdot P_X$, and the hypothesis of interest is whether a test point is drawn from $P_{W|X}$ or from a distinct conditional distribution $Q_{W|X} \neq P_{W|X}$.

The authors propose a new score function based on an estimate of the conditional CDF, which incorporates information from test points to improve estimation accuracy. They also introduce a data-driven method for selecting the kernel bandwidth for each test point to enhance detection power. The authors prove that the proposed method controls the marginal FDR and provide empirical results demonstrating this control and its enhanced power.

**Strengths:**

*   **S1:** The paper introduces a new score function, an estimate for the conditional CDF, which offers a novel approach to the localization problem in conditional novelty detection.
*   **S2:** The paper provides a theoretical result on the convergence rate of the proposed estimate for the conditional CDF, formally justifying the benefit of using test data.
*   **S3:** The p-values are proven to be valid, and the overall procedure provides finite-sample FDR control by utilizing the conditional calibration method (Fithian and Lei (2022), "Conditional calibration for false discovery rate control under dependence").
*   **S4:** The idea of using the test data to improve the estimation of the conditional CDF can be potentially useful beyond the specific application mentioned in this paper.
*   **S5:** The experiments demonstrate FDR control and show improved power compared to baseline methods.

**Weaknesses:**

* **W1:**  The experiments are too simplistic. The paper is framed as a novelty detection work, but the experiments have the flavor of regression. The authors use classic regression datasets and synthetically generated outliers rather than established novelty detection benchmarks. To convince readers of the method's practical utility, a real-world use case is needed. See related question Q5.

* **W2:**  The experimental discussions are limited and lack depth:
     - The evaluation of the data-driven bandwidth selection is limited. It would be very helpful to report the actual bandwidth values selected and illustrate how they vary across different settings. This would emphasize that the choice of bandwidth is non-trivial and highlight cases where it changes in ways that are difficult to predict or set a priori.
     - The experimental settings for the appendix studies (E.1, E.2, E.3) are not specified, it is unclear which data were used.
     - The experiments in Appendices E.1 and E.2 lack sufficient explanation for the observed trends. The authors propose selecting the parameter $w$ based on the accuracy of $\hat{r}$, yet Appendix E.2 neither examines this relationship nor clarifies the results, providing limited practical insight. This leaves no practical guidance for practitioners on how to set $w$ when applying the proposed method. Moreover, the description also does not specify which data were used, leaving it unclear whether $r$ can be computed. I would expect to see an experiment with simulated data (in addition to real-data exp.), where $r$ is known, demonstrating the performance as a function of the accuracy of $\hat{r}$ and different values of $w$.

* **W3:**  The novelty is somewhat limited when viewed in the context of prior work.
     - The main innovation is incorporating test data. The correction using $\hat{r}$ is related to weighted conformal prediction, and the data-driven selection based on the rejection set size is similar in spirit to the method in Marandon et al. (2024), "Adaptive Novelty Detection with False Discovery Rate Guarantee," to whom clearer credit should be given in the main text.
     - The FDR control result is an instantiation of the conditional calibration framework from Fithian and Lei (2022), "Conditional calibration for false discovery rate control under dependence". While the full proof is valuable, this connection should be emphasized before the theorem to properly credit the proof technique.

To conclude, the core idea of using test data to improve the score function is valuable. However, the main reason for my relatively low score (but still positive) is that the authors do not fully convince the reader that this is a practically worthwhile approach. Since the paper's goal is to improve power, this must be demonstrated through more realistic and comprehensive experiments.

**Questions:**

1.  In the simulation details for the real data experiment, the authors mention in lines 661-662 "the null test set is combined with the excluded outliers to construct the complete test set." Does this mean the entire set of outliers was used for each of the 200 repetitions? If so, this choice needs to be justified.
2.  In the novelty detection section (around line 277), $\hat{r}$ is referred to as fixed, but it necessarily depends on the data $\mathcal{D}_c \cup \mathcal{D}_u$. Could you clarify what it is "fixed" with respect to, and whether this affects the validity of Theorem 3?
3.  The computational complexity of the method appears to scale quadratically with the number of test points, $m$. The paper notes a runtime of 30 seconds for sample sizes in the hundreds. What are the practical limitations of this method when $m$ is in the thousands? A brief discussion on scalability should be added.
4. The authors mention that one should choose $w$ based on the accuracy of $\hat{r}$. However, if we have a relatively accurate $\hat{r}$, shouldn't it be reasonable to use it as a score function, as it quantifies the likelihood ratio? The proposed approach is probably more robust, but a short discussion about this would be helpful.
5. In lines 450-451, the authors mentioned "by training a prediction model on the full dataset and take outliers as the 10% of samples with the largest nonconformity scores from both the high-price and low-price groups." Which predictive model is used here? Is it the same one used in the experiments? This should be clarified.
6. Minor comment: There are no details on how the authors estimate $\hat{r}$ (neither in section 4 nor in appendix C). It only appears in Appendix E.1, where it is mentioned that a GLM algorithm was used for estimating r in the main manuscript. This information should be included in section 4 or appendix C.
7.  Minor comment: The sentence in line 916, "both have power reduced with this algorithm," appears to contradict the results in Table 2 for the ALCP method with the quantile score, where power actually increases. Please check and revise this.

---

> ### Author Response · Authors · 2025-11-21
>
> We are grateful for your careful review and insightful comments on our paper. In the rebuttal phase, we have adopted all of your suggestions on both presentation and experiments. Per the weaknesses and questions you raised, we would also like to make some clarifications pointwisely. We kindly refer you to the corresponding parts mentioned in our response in the revised paper for all our modifications (it is available by clicking the pdf icon at the top of the website).
>
> ### Weaknesses
>
> - The preference of using regression settings in our experiments originates mainly from our motivation of testing for conditional outliers. As we mentioned in the introduction, this kind of problems are most common in settings where we want to test for the change of regression relationship between the variable $W$ of interest and covariate $X$. To show the applicability of ALCP in classification settings, we have added a simulated example in Section E.10 (Line 1178) in the revised paper. To summarize, the performance of ALCP under the classification setting is similar to the regression example in the main text. Please see Figure 10 for detailed results.
>
> - Thank you for your suggestions on the experiments. We have performed additional experiments to show the variability of the bandwidth, the influence of $\hat{r}$’s estimation, and its interactive effect with parameter $w$. Generally, the experimental results are aligned with our discussion. The selected bandwidth varies according to the optimal rate with sample size $m$ and conditional variance $\sigma_W$. For density ratio estimation, using an oracle density ratio leads to a larger optimal value of $w$. The additional results are given in Section E.1 (Line 902) and E.4 (Line 979) in the revised paper. Please see Figure 3 and Figure 4 for detailed results.
>   We have also added more explanation for the simulation setting at the beginning of Section E (about Line 897). Hope this is acceptable to you!
>
> - Thank you for your advice. While we have already mentioned the literature of weighted conformal after equation (4), we have added statements about Marandon et al. (2024) at Section 2.2, Line 207 and Fithian and Lei (2022) at Section 3.1, Line 316.
>
> ### Questions
>
> 1. Thank you for your question. We acknowledge the inappropriate setting in the original real-data experiment and have re-performed the analysis accordingly. In the revised version, for each replicate we now randomly sample a subset from the pre-defined outlier set to construct the test data. The corresponding description has also been updated in Section C.
>
> 2. We consider the estimator $\hat{r}$ as fixed here because we obtain it by fitting a binary classifier to distinguish $ \\{V_i \\}_{i \in \mathcal{T}}$ and $\\{V_j\\} _{ j \in \\mathcal{C} \cup \mathcal{U}}$ (in Section 2.2, Line 220). Since the conditional calibration procedure only operates within $D _{\mathcal{C}} \cup D _{\mathcal{U}}$  and the classification algorithm does not distinguish the order of samples, the estimator $\hat{r}$ is fixed with respect to the unordered sets of $\\{V_i\\} _{i \in \mathcal{T}}$ and $\\{V _j\\} _{j \in \mathcal{C} \cup \mathcal{U}\}$. Therefore, it also does not affect the validity of Theorem 3 since we do not break the exchangeability within $D _{c,j}$ for any inlier $j$ when conditioning on $\hat{r}$. Hope this is clear to you!
>
> 3. Thank you for your advice. The comment of a quadratic computational complexity with respect to $m$ is correct, and we have added more discussion in Remark 3, Line 360 about the computational issue. When $m$ is in the thousands, using a single core will take about several minutes. We also note that our method has a clear parallel structure for $j \in \mathcal{U}$ and could be easily parallelized. For even larger $m$, one can reduce the runtime by the number of available cores and keep the overall computation practical.
>
> 4. Thank you for your comment. Using $\hat{r}$ as the score function is feasible for the problem. However, the issue here is that $\hat{r}$ approximates the density ratio of **the inlier distribution** and **a mixture distribution** of many inliers and only a small fraction of outliers. This makes it extremely difficult to estimate $r$ accurately since the “signal” is very weak. Per your advice, we have added Remark 1 at Section 2.2, Line 225 to discuss this issue.
>
> 5. Thank you for pointing this out. We use exactly the same models as in the main experiments: a random forest and a quantile forest, which are employed to compute the residual scores and the quantile regression scores, respectively. We have added a detailed clarification in Section C.
>
> 6–7. Sorry for the confusion. We have added the description of estimating $\hat{r}$ in Section C and revised the explanation at Line 947.

---

> > ### Comment · Reviewer_1Phq · 2025-11-26
> >
> > Thank you for your detailed response and for addressing my comments and questions. I find them appropriately addressed. I maintain my positive score.

---

### Meta-Review · Area_Chair_BkTY · 2025-12-28

**Summary:**

This submission proposes Augmented Localized Conformal p-values (ALCP) for conditional novelty detection by augmenting localized conformal calibration with test-data–assisted kernel estimation and a recalibration step, and it pairs these p-values with BH to control the (marginal) FDR. The reviewers agree the paper is generally well written and the technical components are broadly sound, and they acknowledge empirical evidence of improved detection power under controlled FDR in the presented settings. However, the overall contribution is viewed as incremental relative to localized conformal / conditional calibration frameworks, and the paper does not fully convince the committee of a clear practical impact beyond existing approaches, particularly given the remaining mismatch between the “conditional shift” framing and the (largely marginal) guarantees, as well as concerns about scalability and the breadth/realism of evaluation. After considering the rebuttal and discussion, I recommend rejection.

**Reviewer Concerns:**

The rebuttal meaningfully improves clarity and completeness in several places: it corrects the earlier description around “randomization,” adds citations and positioning relative to closely related prior work, clarifies how the density-ratio model is treated with respect to exchangeability, and expands experimental reporting (including additional studies on bandwidth behavior, estimation quality of the density ratio, and some covariate-shift sensitivity). These changes address a number of presentation and experimental-detail issues raised by the reviewers. That said, the central reservations remain substantially outstanding: the method’s strongest finite-sample guarantees still rely on marginal assumptions (notably requiring matching covariate marginals for finite-sample validity/FDR control), which weakens the conditional-shift motivation and leaves potential vulnerability under covariate shift; the added discussion does not provide a deeper theoretical account of the claimed power boost beyond improved estimation accuracy; and the perceived novelty remains limited because key ingredients (localized conformal ideas, conditional calibration for FDR control, and adaptive selection heuristics) are largely established, with the paper’s main advance being their particular integration and test-set augmentation. Practicality concerns also persist, since the bandwidth-selection procedure can be computationally expensive in m and may be challenging at scale, and the empirical evaluation—although expanded—still does not fully resolve questions about robustness, baseline completeness, and demonstrated real-world utility for novelty detection beyond the chosen case study.

**Reviewer Scores:**

Reviewer 1Phq explicitly stated they “maintain” their positive score after the rebuttal, so I estimate their score remains at 6. Reviewers eRMP and UVEZ did not provide an explicit post-rebuttal score update; given the rebuttal’s clarifications and added experiments, I estimate both would most likely remain at 4, with at most a modest increase to 5 in the optimistic case where the added empirical evidence and clearer positioning sufficiently alleviate their concerns. Overall, the discussion improves confidence in the correctness and presentation, but it does not fully remove the core concerns that drove the below-threshold assessments.

---

### Decision · Program_Chairs · 2026-01-26

Reject